# Networks of genetic similarity reveal non-neutral processes shape strain structure in *Plasmodium falciparum*

Qixin He [1], Shai Pilosof [1], Kathryn E. Tiedje[2], Shazia Ruybal-Pesántez [2], Yael Artzy-Randrup[3], Edward B. Baskerville[1], Karen P. Day[2] & Mercedes Pascual [1,4]

Pathogens compete for hosts through patterns of cross-protection conferred by immune responses to antigens. In *Plasmodium falciparum* malaria, the *var* multigene family encoding for the major blood-stage antigen *Pf*EMP1 has evolved enormous genetic diversity through ectopic recombination and mutation. With 50–60 *var* genes per genome, it is unclear whether immune selection can act as a dominant force in structuring *var* repertoires of local populations. The combinatorial complexity of the *var* system remains beyond the reach of existing strain theory and previous evidence for non-random structure cannot demonstrate immune selection without comparison with neutral models. We develop two neutral models that encompass malaria epidemiology but exclude competitive interactions between parasites. These models, combined with networks of genetic similarity, reveal non-neutral strain structure in both simulated systems and an extensively sampled population in Ghana. The unique population structure we identify underlies the large transmission reservoir characteristic of highly endemic regions in Africa.

[1] Department of Ecology and Evolution, University of Chicago, 1101 E 57th Street, Chicago, IL 60637, USA. [2] School of BioSciences, Bio21 Institute/University of Melbourne, Melbourne, VIC 3010, Australia. [3] Department of Theoretical and Computational Ecology, IBED, and Institute of Advanced Study, University of Amsterdam, Amsterdam, The Netherlands. [4] Santa Fe Institute, Santa Fe, NM 87501, USA. Correspondence and requests for materials should be addressed to M.P. (email: pascualmm@uchicago.edu)

A central question in ecology and evolution regards the extent to which non-neutral processes structure diversity[1–4]. It remains a challenge to identify signature patterns that reveal an important role of ecological interactions in facilitating and stabilizing species coexistence in ecosystems with high diversity, such as tropical rain forests[5,6]. These ecological interactions specifically depend on trait differences between species, so that the assembly of diversity would not simply reflect stochastic colonization and extinction events of equivalent species as under neutrality. Here, we address whether competitive interactions act as a non-neutral stabilizing force that promotes coexistence in another highly diverse system: Plasmodium falciparum populations as an ensemble of diverse strains in regions of high malaria transmission.

Recurrent malaria infections in endemic regions do not generate sterilizing immunity toward subsequent infection[7]; this suggests the existence of a large number of strains of the pathogen. A vast reservoir of P. falciparum exists in local human populations in Africa in the form of asymptomatic infections, hosts that carry the parasite without manifestation of the disease[8]. An understanding of the antigenic diversity of the parasite in such reservoirs, including whether and how this diversity is structured into strains, is fundamental to understanding immunity patterns and developing intervention strategies in the transmission dynamics of P. falciparum malaria.

The high transmission rates of endemic regions suggest frequency-dependent competition among parasites for hosts, through the cross-protection conferred by the adaptive immune system[9,10]. As the success of an infection depends on the immunological memory of a host, new and rare antigenic types have a fitness advantage in the transmission system relative to common ones. In ecological theory, traits that confer such frequency-dependent advantage are known to promote coexistence via the formation of distinct niches, and to oppose in so doing the destabilizing effect of absolute fitness differences in average growth rates[11,12]. The antigenic variation in pathogens maps conceptually to such stabilizing trait differences in ecological competition. Interestingly, "immune selection," a form of balancing selection, is already recognized as an important evolutionary force promoting the diversification and persistence of the var gene family, whose ancient origin predates the speciation of P. falciparum[13,14]. The role of immune selection is much less recognized and understood however for the faster time scales of ecology/epidemiology and for the higher organizational levels of either the repertoires of var genes that constitute a parasite or the population structure of coexisting strains[15].

In high transmission regions, the extensive diversity of the var gene family[16] exhibits low amino acid similarity encoded by different var genes and a very low percentage of genes shared between parasites, both locally and regionally (e.g., < 0.3% in Africa[15,17]). Previous work, known as strain theory, has posited that the non-random association of gene variants within genomes results from selection against recombinants through cross-immunity[18], akin to emergent niches of limiting similarity[19] or selection toward divergent local adaptations[20].

We are therefore interested in addressing whether signatures of immune selection can be detected at the repertoire level in such a diverse system. Existing models for strain theory[18,21,22] do not provide sufficient guidance on expected empirical patterns, because they incorporate so far limited var gene diversity compared to observed numbers for P. falciparum. This is the case for both earlier formulations with a few distinct loci[21,22] and a recent extension for multi-copy gene families[23] that have predicted the emergence of dominant strains with minimum genetic overlap. It is therefore unclear whether population structure can emerge at realistic, high genetic diversity, especially under extensive

recombination rates. Existing strain theory also lacks a neutral counterpart, a neutral hypothesis to disentangle, and statistically test, patterns generated by the acquisition of specific immunity from those resulting simply from the basic demography of the transmission system. In this study, we address these limitations by extending an individual-based stochastic model to incorporate realistic mutation and recombination processes, and generate levels of diversity comparable to those of the var gene family in hyper-endemic malaria regions. In addition, two process-based neutral models are formulated, which include the same epidemiological and evolutionary processes of the full model, but replace specific immunity by either: (i) no immune memory or (ii) generalized protection acquired via the number of previous infections. Last, we propose the application of network properties to identify selection signatures based on comparisons of var repertoire structures under immune selection vs. neutrality. We demonstrate that the structure of genetic similarity networks contains clear signatures of neutral vs. non-neutral processes, and that immune selection plays an important role in shaping the empirical strain structure of var gene repertoires in a local P. falciparum population from Bongo District (BD), Ghana. This structure differs from what is expected on the basis of either previous strain theory[23] or recent ecological theory[24,25], in which niche differentiation takes the form of clusters of strains or species with limited overlap among them. An ensemble of network properties rather than a given clustering metric is therefore needed. We discuss our results in the context of other attempts to identify niche differentiation on the basis of genomic data in microbial ecology[26] and trait differences in community ecology[24,25].

## Results

**Extended individual-based stochastic model of var.** We extended the individual-based stochastic var model of Artzy-Randrup et al.[23] to incorporate more realistic mutation and recombination processes, which allows us to reach levels of diversity comparable to those of the var gene family (Fig. 1; Methods). In the model, each parasite var genome consists of a repertoire of 60 copies of var genes (Fig. 1a). Each var gene is considered a linear combination of two epitopes based on the empirical description of two hypervariable regions in the var tag region of the DBLα domain[27]. The transmission system is composed of a pool of these gene variants and a local human population open to immigrant parasites, in which we track transmission and infections. Simulations start with a static pool of var genes consisting of random combinations of the two epitopes (Fig. 1b). During transmission events, mitotic recombination and mutation generate new var genes, making the overall pool of epitopes effectively open to innovation, whereas meiotic recombination shuffles the composition of var genes of two or more repertoires in co-infections during the vector stage of transmission (Fig. 1c). The epitopes in the genes represent components of the PfEMP1 molecule that are recognized and remembered by the immune system of the host; they are the 'traits' that effectively mediate competition for hosts at the population level.

In the immune selection model, individuals gain protection against specific epitope variants, through expression of the corresponding genes in an infection. Therefore, the rate at which hosts clear the infections increases with a higher number of specific epitopes seen from past infections. We then developed two neutral models and compared the repertoire structures they generate with those of the model with specific immunity. The first neutral model assumes generalized immunity, in which protection is acquired as a function of the number of previous exposures irrespective of their specific antigenic identity; the second one is a

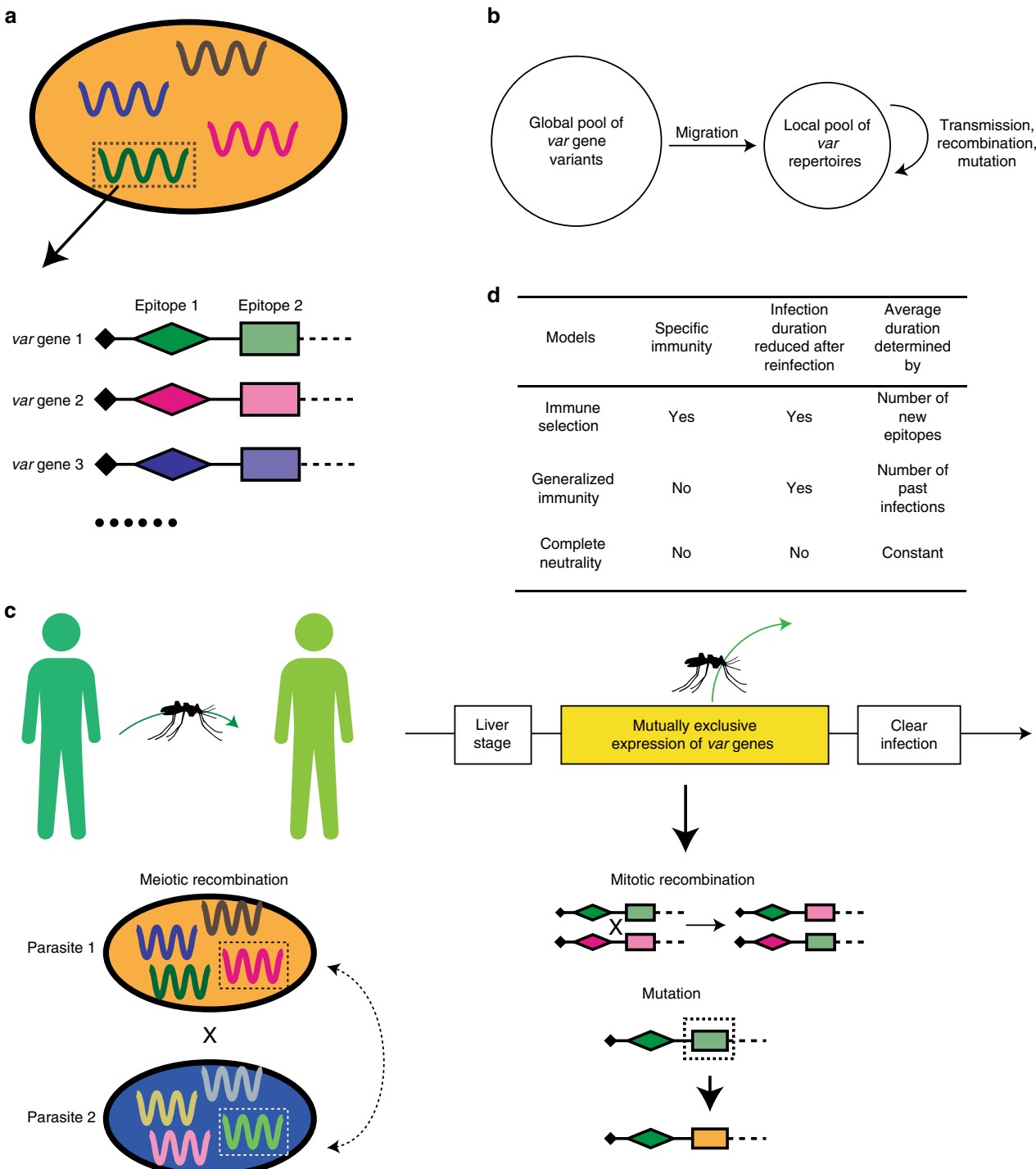

**Fig. 1** Schematic illustration of the *var* gene model. **a** Each parasite genome (ovals) consists of a repertoire of *g* copies of *var* genes. Each *var* gene (depicted by different colors within each parasite) is in turn represented as a linear combination of epitopes (depicted by different shapes), with each epitope having many possible variants (alleles, depicted by different colors). **b** The local population receives *var* repertoires from a fixed global *var* gene pool through migration. **c** At each transmission event, one donor and one receiver host are selected at random from the host pool. Each parasite genome in the donor host is transmitted to the mosquito with probability of 1/(number of genomes). During the sexual stage of the parasite (within mosquitoes), different parasite genomes can exchange *var* repertoires through meiotic recombination to generate novel recombinant repertoires. The receiver host can receive either recombinant genomes or original genomes. During the asexual reproduction stage of the parasite (within the blood stage of infection), *var* genes within the same genome exchange epitope alleles through mitotic (ectopic) recombination. Also, epitopes can mutate. These two processes generate new *var* genes. Each *var* gene is expressed sequentially and the infection ends when all the *var* genes in the repertoires have been expressed. A new transmission event may occur throughout the period of expression of *var* genes as the result of biting events. **d** Comparison of the three different models. Only the immune selection model includes specific immunity, through the dependence of infection duration on the memory of previous alleles that have been seen by a given host. For meaningful comparisons, all the models have the same mean duration of infection

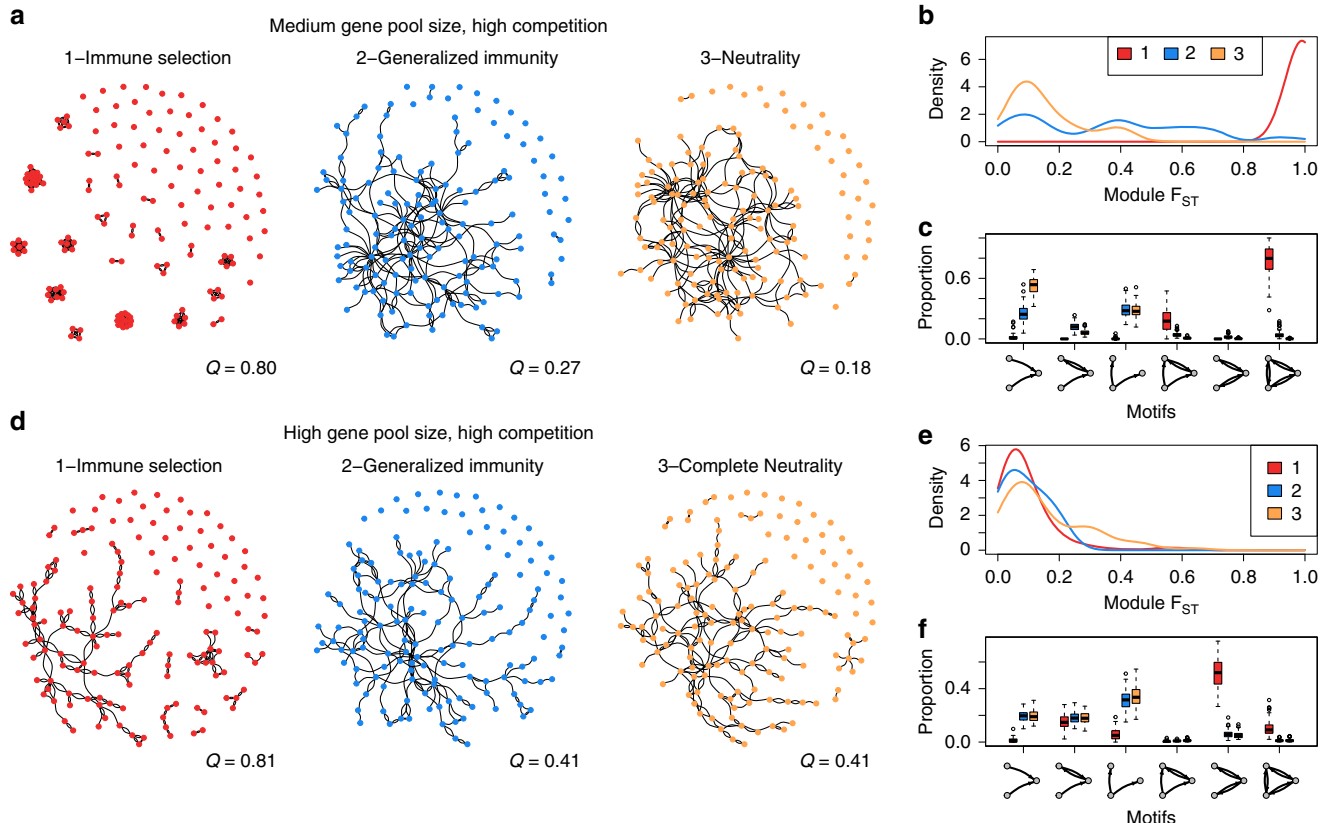

**Fig. 2** Population structure for the three different models quantified using network properties. Shown here are repertoire similarity networks and representative network metrics across scenarios under different diversity regimes generated from model simulations with high competition (high duration of naive infection and high transmission rates). Upper panel, medium diversity (gene pool size of 1,200) and lower panel, high diversity (gene pool size of 24,000). **a**, **d** Comparisons of repertoire similarity networks of 150 randomly sampled parasite *var* repertoires from one time point under three scenarios. Edge width is relative to the strength of genetic similarity between pairs of repertoires. Only the top 1% of edges are drawn and used in the analysis (see Supplementary Fig. 3 for distribution of edge weights). Within the largest component of each network, the size of each repertoire is relative to its normalized betweenness centrality. The value of maximum modularity $Q$ is calculated using edge betweenness[30] and is shown at the lower right corner of each network. The modules obtained in these networks represent groups of highly similar repertoires (strain modules), which are conceptually similar to geographically isolated populations with limited gene flow. We therefore calculate the pairwise $F_{ST}$ of strain modules, to quantify how different strain modules are from each other, providing a measure of limiting similarity that compares within-module and between-module diversity[64]. **b**, **e** Pairwise module $F_{ST}$ distributions of 100 repertoire similarity networks per scenario. **c**, **f** Distributions of the proportion of occurrences of three-node graph motifs for the three models across 100 repertoire similarity networks. The box shows the interquartile range (IQR, from the 25th to the 75th percentile of the distribution), and the lower and upper whiskers correspond respectively to 1.5 IQR of this lower quartile and 1.5 IQR of the upper one, with the median indicated with a line and points displaying outliers

completely neutral model in which infections propagate and recover, but hosts are blind to any history of exposure so that repertoires do not compete for hosts (Fig. 1d). These two neutral models include all the epidemiological processes, except for specific immunity towards the *var* genes that a host has been exposed to, and the resulting cross-protection. The epidemiological phenotype under immune selection is the duration of naive infection ($D$). The parameter, $D$, directly influences the basic reproduction number (or fitness) of the parasite, $R_0$, and thus we match the infection period in the complete-neutrality model to the average duration that emerges in the corresponding immune selection model for each specific set of parameters (Fig. 1d; Methods). Similarly, we match the distribution of infection duration in the generalized immunity model to the emergent distribution of the full model. Therefore, repertoires under these neutral models do not exhibit fitness differences related to their specific genetic composition, whereas repertoires under the specific immunity model do differ in fitness as a result of the aggregated history of infection of the host population

(Supplementary Fig. 1). The resulting Entomological Inoculation Rate (EIR), a measure of the force of infection experienced by individual human hosts measured as the number of infectious bites per host per year, is comparable among the three models. However, as the complete neutrality model does not include any mechanism to generate the typical empirical age infection distribution with higher prevalence of asymptomatic infection in children[8], prevalence in this model tends to be higher than that in the corresponding immune selection and generalized immunity models (Supplementary Fig. 2).

**A network representation of var repertoire diversity**. The reticulate evolutionary pattern of *var* genes, generated by frequent mitotic and meiotic recombination within and between parasite genomes[28,29], respectively, precludes the application of traditional population genetics tests for balancing selection (e.g., Tajima's $D$). Hence, we develop an application of network theory to study the evolution of *var* repertoire structures and show that the structure of genetic similarity networks contains clear signatures of neutral

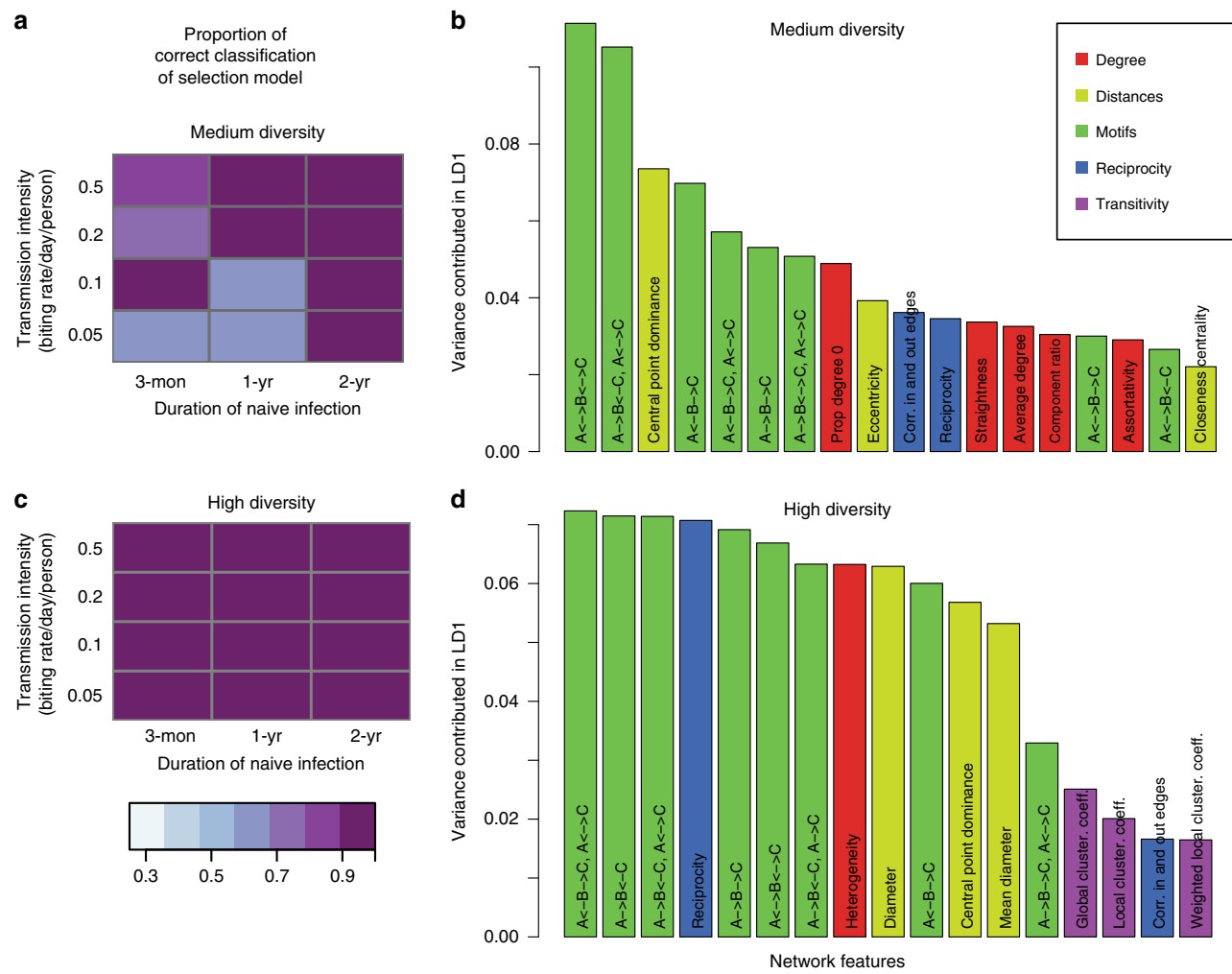

**Fig. 3** Importance of network features for the classification and the power of correct classification of the selection model. Results are for different levels of duration of naïve infection, biting rate, and *var* gene pool size (medium diversity [1,200–2,400] in **a**, **b** and high diversity [12,000–24,000] in **c**, **d**). **a**, **c** The shade of colored squares indicates the proportion of correct assignments of the immune selection scenario. Variance contribution of network features in the first linear discriminant function (LD1) are displayed with color-codes corresponding to the feature groups (**b**, **d**; see Supplementary Table 1 for feature groups). Only top features that explain at least 90% of the variance are shown. In simulations with a gene pool of medium diversity **a**, the proportion of correct assignments of the selection model increases with increasing infection duration and biting rate. When the genetic pool is of high diversity, the selection model is almost always perfectly assigned **c**, while neutral and generalized immunity models are harder to differentiate, even under high transmission and long infection durations (see Supplementary Table 2). The proportions of different motifs are the most important features that discriminate scenarios in both high and medium diversity. In the high diversity scenarios, there are a higher number of important features that equally contribute to the classification **d**, as compared with the medium diversity scenarios **b**. The classifiers shown in this figure are built with the top 10% edge weights of the repertoire similarity matrices

vs. non-neutral processes. We analyzed the genetic structure of the parasite population using networks in which nodes are *var* repertoires, weighted edges encode the degree of overlap between the epitopes of these repertoires, and the direction of an edge indicates whether one repertoire can outcompete the other (Methods). Comparisons of structure across the similarity networks generated under the three models reveal distinctive features of immune selection, although the specific features that distinguish immune selection vary under different epidemiological settings (described below).

As *var* genes exhibit different diversity levels across different endemic regions[17], we investigated the influence of *var* gene pool size (i.e., the number of *var* genes in the global pool; Fig. 1b) on the immune selection signatures. As the two most relevant epidemiological parameters, transmission intensity and duration of a naive infection, determine the intensity of competition among *var* repertoires, we vary them systematically to address

their influence on signatures of immune selection. Higher transmission and longer duration of a naive infection intensify competition among repertoires. They also increase the rate of meiotic recombination among repertoires in mosquitoes. It follows that signature patterns of immune selection should be most evident with increasing values of duration, gene pool size and transmission, for conditions representative of high endemicity.

To explore selection signatures in networks generated under different intensity of competition between strains, we use a suite of network metrics (see complete list in Supplementary Table 1 and see Supplementary Fig. 3 for the low competition scenarios). If a process akin to limiting similarity underlies population structure, networks are expected to be partitioned into disconnected clusters of highly similar repertoires that occupy separate niches in antigenic/genetic space. One way to quantify the partitioning of a network is by calculating maximum modularity

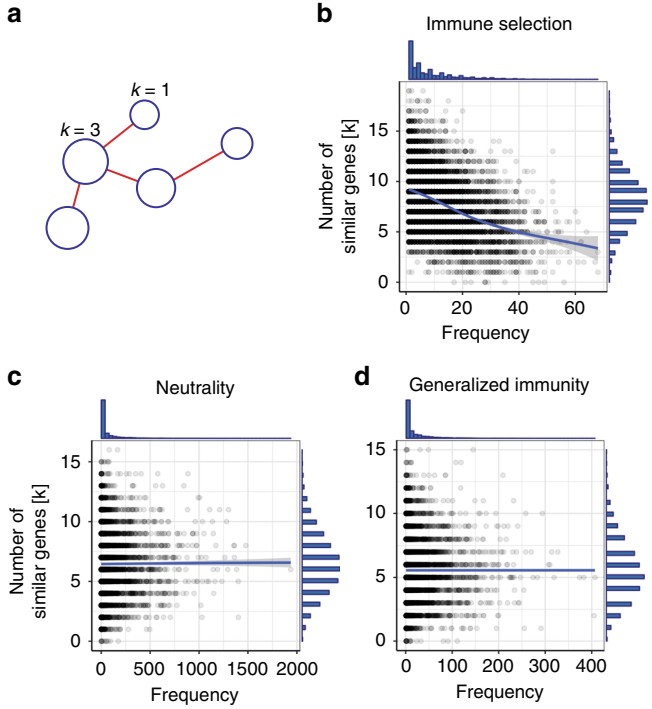

**Fig. 4** The relationship between gene frequencies and number of genes similar to a focal gene is different between selection and null models. **a** In a gene similarity network, each node represents a unique gene circulating in the population and the edges encode the sharing of at least one allele between genes. The size of the node is proportional to its frequency in the population and the node degree $k$ depicts the number of genes that share at least one allele with the focal gene. There is a negative correlation between a gene's frequency and degree in the immune selection scenario (**b**, $r = -0.412$, $t$-test: $p$-value $< 2.2e - 16$), but not in the complete neutrality (**c**, $r = 0.016$, $t$-test: $p$-value $= 0.17$) or generalized immunity (**d**, $r = -0.002$, $t$-test: $p$-value $= 0.89$) scenarios. $G = 24{,}000$, $b = 0.5$, $D = 1$ year

$(Q)$[30]. Under high competition, when the local *var* gene pool is of a medium size characteristic of endemic regions of Asia/Pacific (~ 1200–2400 different *var* genes)[31], the selection case differs notably from those of the two neutral models: repertoires are typically grouped into well-defined modules (expressed as high $Q$ and module $F_{ST}$ values, Fig. 2a, b), whereas in networks resulting from the two neutral models, nodes are typically connected to form star-shaped or tree-like structures. This qualitative difference in structure resembles the prediction of non-overlapping strains in classic strain theory, where the disconnected clusters are analogous to niches in immune space consisting of highly similar repertoires.

In addition, because competition at the repertoire level in the selection case promotes matching competitiveness of two given connected repertoires, it results in reciprocally connected directed edges of similar weights. In contrast, in the two neutral models, repertoires with lower number of unique genes are not removed by selection and, when one repertoire outcompetes another, there is only one directed edge between the pair. We use three-node motifs to capture this variation in competitiveness. For example, a binary in-tree motif (A- > B < -C) reflects that repertoire B is outcompeted by A and C, whereas a complete graph motif in which three repertoires are all reciprocally connected (A < - > B < - > C < - > A) indicates a balanced, reciprocal competition. We find that networks of the selection model have a high proportion

of such reciprocal motifs compared with those of the two neutral models. Binary in-tree or out-tree motifs are instead the most common in the neutral models, reflecting parent–offspring evolutionary relationship, resulting from recombination where the recombinants are not purged by immune selection (Fig. 2c, f).

Under a regime with a larger initial gene pool that matches the diversity levels of endemic regions in Africa (~ 12,000–24,000 different *var* genes)[15], repertoires have a lower genetic overlap compared with medium-size gene pool (see Supplementary Fig. 4). This pattern follows naturally from increased gene pool diversity, because repertoires can be formed from a larger number of gene combinations. Although such low overlap can indicate a non-random structure[15], it cannot per se distinguish selection from the two neutral models. Accordingly, module $F_{ST}$ is low in all three cases and is not a good indicator of selection (Fig. 2e), despite the selection model possessing more separate components than neutral models (Fig. 2d). Nonetheless, networks generated with immune selection can still be differentiated from those generated under neutral models using motif composition (Fig. 2f), as well as other network metrics (see Fig. 3 and Supplementary Fig. 3). In particular, the weaker similarity between repertoires entails a less clear network partitioning than that of lower gene pool diversity systems. We therefore use betweenness centrality as a property reflecting their limiting similarity: this metric measures the importance of a repertoire in a network by calculating the proportion of shortest paths connecting any pair of nodes that go through it[32]. For the networks generated with the neutral models, betweenness centrality varies little among repertoires, with no highly central ones (Fig. 2d). This is because the persistence of each repertoire is independent of the antigenic composition of other repertoires given the lack of specific competition. By contrast, in the selection case some repertoires are clearly more central than others (Fig. 2d), reflecting the non-random persistence of antigenic niches, connected through these hubs via a series of recombination events.

**Network classification at the repertoire and gene levels**. To apply the findings from network theory to empirical data, we first asked whether networks produced with the agent-based model can be classified into the processes that generate them—immune selection vs. generalized immunity or complete neutrality—using an ensemble of network properties (Supplementary Table 1). As competition is mainly among repertoires that are highly similar, we inspected whether the quantile of edges ranked by similarity values that are left in the networks influences the accuracy of classification. We found that the power of classification (measured by the proportion of correct classifications [true positives and true negatives]; Methods) remains largely unaltered as long as the bottom 20% in similarities of edges are removed (see supplementary Table 2). With a medium gene pool size, there is a positive correlation between the transmission intensity and our ability to classify networks correctly, reflecting an increasing divergence between the different kinds of networks (Fig. 3a). With a high gene pool size, the classification always differentiates the selection scenario from the two neutral ones correctly (Fig. 3c). The proportion of different kinds of motif structures is the most powerful metric for the classification in high and medium diversities. Although metrics related to node degree contribute more in classification under medium gene pool size, metrics based on reciprocity and distance are more important for high gene pool size. Overall, a higher number of network features contributes evenly to the classification under the high diversity scenario than under the medium diversity one (Fig. 3b, d).

Most endemic regions have seasonal transmission of malaria. Therefore, we repeated the classification analysis using networks

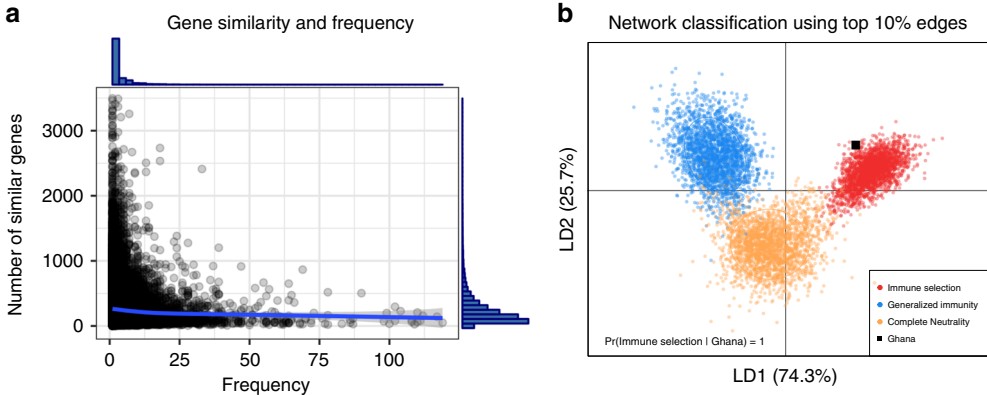

**Fig. 5** Empirical investigation of the Ghana data shows resemblance to the immune selection scenario. **a** Negative correlation between DBLα type frequencies and their number of similar genes for the upsB/upsC *var* genes in the parasite population ($r = -0.040$, $p < 2.2\text{e} - 16$). The number of similar genes is calculated as the degree ($k$) of a focal gene in the gene similarity network, for amino acid similarities above 0.6. Histograms on the top and right of the plot show the distributions of $k$ and DBLα-type frequencies, respectively. **b** Classification of networks generated with the agent-based model using Discriminant Analyses of Principle Components[66] onto a 2-D space formed by the two linear discriminant (LD) functions using the top 10% edge weights. The empirical network is more likely to be generated under an immune selection regime (posterior probability [PP] = 1), as opposed to neutrality (PP = 3.57E – 8) or generalized immunity (PP = 2.15E – 9). The classification relies on comparisons of 34 network properties (see Supplementary Table 1) trained with 7000 simulated networks and verified on test sets of 800 networks (i.e., 100 combinations of different parameter settings for each of the scenarios were run; infected hosts were sampled in October and the next June each year, as for the empirical sampling, at the stationary stage of the simulations (i.e., the last 26 years in the simulations); similarity networks were then built for randomly sampled parasites from the sampled hosts). Accuracy of network classifications is above 0.99 for each scenario (see Supplementary Table 2 for comparisons of accuracy and the classifications of the empirical network using different percentage of top edges in the network, and Supplementary Fig. 5 for motif properties of the empirical network compared with simulated ones)

generated with the agent-based model that includes a seasonal transmission pattern with a high gene pool size (Methods). Here again, we found that classification power remains largely unaltered as long as the bottom 20% of links are removed. Unlike in non-seasonal networks, the classification can also differentiate generalized immunity from complete neutrality scenarios (Supplementary Table 2), due to the differences in the distribution of duration of infections (see Supplementary Fig. 1). Specifically, in the scenario of generalized immunity, early infections in naïve hosts persist long enough to overcome the bottleneck in transmission during the dry season and are therefore carried on into the wet season. In contrast, infections under complete neutrality have relatively short durations that limit persistence through the dry season. Therefore, strain composition has a faster turnover rate under complete neutrality than under generalized immunity. These different dynamics leave different signatures in the networks, including motif compositions (Supplementary Fig. 5).

In addition to the effects of immune selection at the repertoire level, frequency-dependent competition also works at the level of *var* genes in the population. Specifically, frequency-dependent competition will limit the abundance of genes that are similar to many others in their epitope composition—and are thus readily recognized by the immune system—while favoring the abundance of genes with a unique composition of epitope alleles. We can test this prediction using a network in which nodes are genes and edges encode similarity in allelic composition (Fig. 4a). In the immune selection case, we find a characteristic negative correlation between node degree (number of genes similar to a focal gene) and the frequency of genes in the host population, because only genes that occupy a different niche can reach high frequencies. This effect is absent in the neutral models (Fig. 4b–d).

**Comparison with empirical data**. Deep genetic sampling of local populations in BD, Ghana allows application of these theoretical

findings to examine the role of immune selection in nature. Gene similarity networks were built from *var* DBLα domain tags sequenced from 1248 *P. falciparum* isolates in Ghana (Methods). An isolate refers to a complete sample of parasites from a host, which may contain multiple infections (i.e., multiplicity of infection, or MOI > 1). We restrict our analyses to the upsB/upsC group of the DBLα domain, because this subset is known to exhibit frequent ectopic recombination within itself relative to the more conserved upsA group[29]. This subset is therefore less prone to generate the above negative correlation spuriously out of differences in recombination rates, and it provides a more appropriate counterpart to our theory, which does not consider functional differences between *var* gene variants. The resulting gene similarity network exhibits a strong negative correlation between *var* DBLα-type frequency and number of similar neighbors, providing evidence for frequency-dependent competition (Fig. 5a).

We then examine the similarity network at the level of repertoires, by calculating shared DBLα types between different repertoires in a subsample of 161 isolates whose MOI is equal to one (i.e., whose *var* genes most likely compose a single repertoire; Methods; Supplementary Fig. 5). We applied our network classification method to ask whether immune selection had an important role in shaping empirical population structure. We generated a library of simulated networks under parameter ranges representative of the sampling area. In order to represent seasonality in the empirical data correctly, we ran the simulations using a seasonal biting rate that is characteristic to the local area based on Appawu et al.[33] (Methods). The resulting values of EIR [~ 25–170] encompass empirical observations for the region[33]. We sampled 94 and 67 parasites every October and the next June, respectively, to construct the similarity networks that represent the empirical sampling (Methods). Classification of the empirical network obtained with the top 10% of edges in similarities, and therefore the potentially strongly interacting repertoires, indicate its resemblance to networks generated with the immune selection scenario based on discriminant functions (Fig. 5b). This

classification result is robust up to a similarity strength cutoff value of the top 25% of the edges. Beyond this point, when weaker edges are considered, results assign the classification to generalized immunity (Supplementary Fig. 6). The lower 75% edge weights in the empirical networks represent, however, the sharing of at most one gene between pairs of repertoires, which provides little information on strongly interacting repertoires. This is discussed below.

## Discussion

The findings provide clear evidence for the role of frequency-dependent competition as a result of immune selection in structuring antigen composition in a natural population of *P. falciparum*. Network patterns of repertoire similarity differed significantly from those obtained by the demography of transmission alone. The importance of immune selection at the *var* gene level has been recognized before, based on the ancient origin[13,14] and sequence diversity of the gene family itself[16]. An agglutination experiment by Bull et al.[34] also showed that children who have broader protection against *Pf*EMP1 variants tend to be infected by only rare variants. Our results provide an additional piece of evidence of selection at the gene level, in that common *var* gene variants can only persist if they are genetically distinct. That immune selection operates at the level of the genes does not necessarily imply an effect at the higher level of organization of repertoires or strains, especially given large gene pools and high recombination rates. We have shown that under extreme genetic diversity and intense recombination, immune selection profoundly structures repertoire diversity, and that this structure can be detected and characterized using network metrics.

The network signatures we have identified present conceptual similarity to traditional tests of balancing selection developed in population genetics or community ecology (as summarized in Supplementary Table 3), thus filling the gap of available tests for highly recombinant gene families. Specifically, a higher number of components/communities with even sizes is observed under the immune selection scenario, in analogy to evenness measures in ecology or Tajima's *D* in population genetics. Another key pattern is limiting similarity. It is akin to high linkage disequilibrium (i.e., non-random association) among genes under balancing selection, such as in HLA genes[35]. In ecology, early competition models such as the standard Lotka–Volterra equations showed that stable coexistence requires a given degree of niche divergence[19]. The limit of coexistence is now understood to depend on both stabilizing (frequency-dependent) niche differences and destabilizing (fixed) fitness differences in average growth rates[11]. Moreover, more recent theory on niche differentiation has described the emergence of clusters of similar species in trait space, where the limited overlap is now between these groups and not individual species[24]. This pattern is a long-lived transient state, which can be stabilized when immigration[25] or evolution[24] are introduced explicitly. It is the basis for the gap statistics method to distinguish simulated communities under niche differentiation from null models based on two clustering measures, *k*-means dispersion[36] and Ripley's *K* function[37]. Clustering on the basis of genomic data was also applied to this end in microbiome communities[26], although limitations of the null model confound the results[38]. The coexistence pattern of highly similar and divergent species is analogous to the population structure we find in our model under a low to medium diversity of the gene pool, consistent with previous studies[23], as well as the genetic patterns resulting from balancing selection in evolutionary genetics (Supplementary Table 3). In that regime, here too, competition-driven clustering is identified by our network metrics. In

particular, modularity detects communities that are more highly linked within than between; module $F_{ST}$, introduced here, quantifies linkage based on genetic distinctiveness of a focal group weighted by its genetic diversity.

Importantly, clustering is no longer a characteristic property in our system at high diversity of the gene pool, comparable to that observed in endemic regions. Thus, cluster measures alone are unable to differentiate immune selection from neutral processes and we suspect the same will be the case in ecological systems if trait spaces underlying interactions are high-dimensional[39]. Existing models in community ecology have considered so far only a low number of dimensions.

For the *var* system, we have therefore relied on an ensemble of network properties. Motif compositions provide a detailed profile of similarity patterns among triplets, which were found to explain the largest proportion of the variance between the selection and neutral models (Fig. 3). In addition, reciprocity in the network captures a unique signature of balancing selection in multi-copy gene families, which does not have a counterpart in ecology or population genetics measures. This is because the fitness of repertoires depends on the number of unique genes under immune selection, given functional equivalence between individual genes. Competition among genomes not only selects for those with diverse genes, but also for the ones with more genes. Accordingly, genomes in West Africa, where competition is intense, maintain more than 50 unique *var* genes, whereas those in South America, where transmission and therefore competition is low, are 10 *var* genes shorter[40]. More generally, these network properties can be applied to other multicopy gene repertoires for antigenic variation, such as *vsg* genes in *Trypanosoma brucei* or *msg* genes in *Pneumocystis carinii*[41].

Detecting signatures of selection requires a comparison to dynamic neutral models. The observation of high modularity or low pairwise similarity among repertoires does not guarantee the importance of immune selection, as low gene pool size or low transmission can also produce similar patterns under neutral models (see network structures in Supplementary Fig. 3). It is likely that generalized immunity, neutrality, and immune selection are all at work, either together or at different stages or genes. We have classified networks to these three scenarios separately to investigate whether a distinct population structure exists for *var* repertoires that reflects an important role of immune selection. In other pathogen systems, the role of neutral vs. non-neutral forces were also investigated, which revealed unforeseen patterns. For example, Nicoli et al.[42] found that strain replacement is more likely under generalized immunity than strain-specific immunity when vaccination is applied in Pertussis. Cobey and Lipsitch[43] demonstrated that weak specific immunity together with generalized immunity permit the coexistence of strains with weak competence in *Streptococcus pneumoniae*.

Another interesting feature of the *var* repertoire dynamics is the positive feedback between increasing competition and the generation of inferior competitors. In endemic regions with higher rates of transmission, hosts exhibit higher average MOI[44]. The rate of generation of recombinant genomes is positively correlated with MOI, because mosquitoes are more likely to pick up multiple *var* repertoires from the host. As unfit repertoires are generated at a higher rate under higher transmission, immune selection leaves stronger signatures in high transmission regimes, compared with lower transmission (Figs. 2, 3). Conversely, ectopic recombination can create new antigenic variants, conferring a fitness advantage to strains with rare variants. Therefore, in areas of high transmission, there is a high generation of allelic diversity, which is maintained by selection. These two forms of recombination, caused by different molecular mechanisms, result in different directions of selection.

We have explored parameter space broadly in terms of transmission intensity and gene pool size, and have also considered seasonal and non-seasonal transmission. Additional variations of model structure will be addressed in future work. In particular, the rules that represent within-host dynamics and regulate the expression of multiple infections and their interactions with host immune responses can be examined in more detail. The genetic multiplicity of *P. falciparum* was found to potentially prolong asexual infections[45,46], probably due to cross-reactive immune responses[47,48]. Although for our main results, we did not consider a cost associated with higher MOI, the introduction of a more prolonged infection within multiply infected hosts and across a range of parameters, indicated the robustness of the described network structures.

Another extension of the model should relax the functional equivalence of all *var* genes[9,49–51]. In the empirical network analysis, when edges are included that represent the lower part of the similarity distribution (the 75% weakest edge weights, corresponding to links between repertoire pairs that only share one gene), the empirical pattern resembles that obtained under generalized immunity (Supplementary Fig. 6). As the majority of the one-gene sharing edges involves the most common *var* types, the pattern could arise from these genes having more critical functions than other *var* types, violating model assumptions of functional equivalence. The pattern could also arise from differential selection pressure on the genes. Although our model assumes that parasites express all *var* genes in the repertoire during the infection, ensuring equal selection pressure on all the genes, empirical experiments have shown evidence of a structured switching pattern, in which slow switching *var* genes express more prominently than fast switching ones, preventing exhaustion of the repertoire in one infection[52–55]. Thus, the activation hierarchy could bias *var* genes toward differential strength of immune selection. Other open areas for further investigation of *var* evolution include phenotypic mapping of sequence diversity to immunity, population structure over time, and its influence on responses of the malaria system to interventions.

Early motivation for strain theory was the recognition that organization of *Pf*EMP1 variants (and their underlying genes) into persistent largely non-overlapping sets can deeply alter our understanding of epidemiology and control, e.g., by viewing *P. falciparum*'s apparent large reproductive number ($R_0$) as resulting from a large ensemble of strains with much lower reproductive numbers[21]. With the sheer number of existing and ever-changing variants and repertoires, previous definitions of strains as long-lived entities do not apply at high endemicity. The resulting population structure nevertheless exhibits limited similarity, in the form of sparse small clusters and/or isolated individual repertoires interspersed into voids in antigenic/genetic space, instead of well-defined niches. This emergent structure provides an image of competition at the "limit" of limiting similarity because of immense diversity. The resulting coexistence and diversity at the different hierarchical levels of genes and repertoires would enable the large reservoir of asymptomatic infections that makes malaria so resilient to elimination in high transmission regions. As such, monitoring *var* gene diversity and structure in responses to control efforts becomes central to understanding malaria epidemiology and to creating a theory of control grounded in the reality of the complexity of the system.

## Methods

**The extended *var* evolution model**. Model parameters and symbols are summarized in Supplementary Table 4. The diversity of *var* genes is represented at three organizational levels corresponding respectively to alleles (epitopes), genes, and repertoires. Specifically, each parasite genome consists of *g* *var* genes (in the main text, *g* = 60). The specific combination of the *var* genes is referred to as a *var*

repertoire throughout the paper. Each *var* gene is in turn a linear combination of *l* loci (in the main text, *l* = 2) encoding epitopes that are connected linearly and each epitope can be viewed as a multi-allele locus with *n* possible alleles. Immune selection in the model is a result of specific immunity to epitope variants (alleles), which represent components of the *Pf*EMP1 molecule that are recognized and remembered by the immune system of the host[56]. In effect, these epitopes serve as traits that mediate competition for hosts at the population level, as individuals gain protection against specific alleles expressed by the parasite during an infection (Fig. 1a).

Initiation of the simulation: To initiate the *var* gene pool G, a random allele for each epitope is chosen from the *n* alleles to form each gene (Supplementary Table 4). In the simplest case, if there are two epitopes in a *var*, then a particular *var* $g_i = \{L_{i1}, L_{i2}\}$, where $L_{i1}, L_{i2}$ are random numbers from $U(1, n)$. With $n_i$ possible alleles at each epitope, the total number of possible genes is $\prod n_i$. However, we chose G at least five times smaller than $\prod n_i$ so that not all combinations of alleles constituting a gene will be available. This choice is based on the fact that not all combinations of alleles form viable proteins. In the beginning of the simulation run, 20 hosts were selected and infected with distinctive parasite genomes that consist of sets of *g* *var* genes randomly selected from the pool G. The size of the host population, H, is kept at a constant size (i.e., when a host dies, a new host is born). For the age structure in the simulations, we fitted an exponential distribution to the reported Bongo population demography and estimated an average life span of 30 years. Therefore, each host has a death rate of d = 1/30 per year.

Repertoire transmission: Vectors (mosquitoes) are not explicitly modeled. Instead, we set a biting rate *b* so that the average waiting time to the next biting event is equal to $1/(b \times H)$. The force of infection is kept the same across host age in the model. When a biting event occurs, two hosts are randomly selected, one donor and one recipient. If the donor has infectious parasite repertoires and the receiver is infected with a probability of *p* (i.e., transmission probability). If the donor is infected with multiple strains in the blood stage, then the transmission probability of each strain is *p* divided by the number of infectious repertoires. A delay is applied between a transmission event and a repertoire becoming infectious to account for stages of the life cycle in the vector and in the human host that are not modeled explicitly (see below).

Meiotic recombination: Meiotic recombination occurs between strains in the sexual stage of the parasite's life cycle. When multiple strains are transmitted to the donor, these strains have a $(1 – P_r)$ probability to remain as the original strain and a $P_r$ probability to become a recombinant strain, with $P_r$ calculated as follows,

$$P_r = 1 - 1/N_s \qquad (1)$$

where $N_s$ is the number of strains transmitted to the donor. Although the association of physical locations and major groups of *var* genes is established, orthologous gene pairs between two strains are often unknown. Therefore, we implement recombination between strains as a process in which *g* genes are randomly selected out of all the original genes from the two strains pooled together. As physical locations of *var* genes can be mobile, this assumption is a reasonable simplification of the meiotic recombination process.

Ectopic recombination within the strain in the asexual blood stage: *Var* genes often change their physical locations through ectopic recombination and gene conversions. These processes occur at both sexual and asexual stages. However, ectopic recombination is observed more often in the asexual stage, where the parasites spend most of their life cycle[6]. Therefore, we only model ectopic recombination among genes within the same genome during the asexual stage. Two genes are first selected from the repertoire. Then, the location of the recombination breakpoint is randomly chosen, so that loci to the right of the breakpoint are either swapped (recombination) or copied (gene conversion). In the current implementation, we assume all breakpoints result into recombination rather than gene conversion. Finally, newly recombined genes have a probability $P_f$ to be functional (i.e., viable) defined by the similarity of the parental genes:

$$P_f(x) = \tau^{\frac{x(\delta-x)}{\delta-1}} \qquad (2)$$

(Eq. 3 in Drummond et al.[57]), where *x* is the number of mutations between the recombined gene and one of the parental genes, *δ* is the genetic difference between the two parental genes, and *τ* is the recombinational tolerance. If the recombined gene is selected to be non-functional, then the parental gene will be kept. Otherwise, the recombined gene will substitute the parental gene so that a repertoire with a new gene is formed.

Mutation: Mutations occur at the level of epitopes. While infecting a host, each epitope has a rate of mutation, *μ*, to mutate to a new allele so that *n* increases by one. New mutations can die from lack of new transmissions, proliferate through new transmissions of the repertoire within which they mutated, incorporate into other genes through ectopic recombination, or be transferred to a different repertoire through meiotic recombination.

Within-host dynamics: Each strain is individually tracked through its entire life cycle, encompassing the liver stage, asexual blood stage, and the transmission and sexual stages. As we do not explicitly model mosquitoes, we delay the expression of each strain in the receiver host by 14 days to account for the time required for the sexual stage in the mosquito and the liver stage. Specifically, the infection of the

host is delayed 7 days to account for the time required for gametocytes to develop into sporozoites within mosquitoes. When a host is infected, the parasite remains in the liver stage for an additional 7 days[58] before being released as merozoites into the bloodstream, invading red blood cells and starting the expression of the *var* repertoire. The expression of genes in the repertoire is sequential and the infection ends when the whole repertoire is depleted. During the expression of the repertoire, the host is considered infectious with the active strain. The expression order of the repertoire is randomized for each infection, whereas the deactivation rates of the genes are controlled by host immunity. When one gene is actively expressed, host immunity "checks" whether it has seen any epitopes in the infection history. In the immune selection model, the deactivation rate changes so that the duration of the active period of a gene is proportional to the number of unseen epitopes. Duration of infection is not varied a priori as a function of age but is instead determined by whether a given host has seen the particular strain in the past. Therefore, the duration of infection of a particular repertoire in a particular host is,

$$\text{Total duration} = \frac{D}{g} \times \sum_{j=1}^{g}(\text{No. new epitopes})/l \qquad (3)$$

After the gene is deactivated, the host adds the deactivated gene alleles to their immunity memory. A new gene from the repertoire then becomes immediately active and the strain is cleared from the host when the whole repertoire of *var* genes is depleted. The immunity toward a certain epitope wanes at a rate $w = 1/100$ per day[59]. In the current implementation, we assume no cost associated with MOI, i.e., the duration of infection is not correlated with the number of genomes a host is infected with (blue line). We also explored another version of within-host dynamics, in which the duration of infection increases as a function of MOI. The two scenarios of within-host dynamics produce qualitatively similar network structures (see Supplementary Fig. 7).

Implementation of the simulation: The simulation is an individual-based, discrete-event, continuous-time stochastic model in which all known possible future events are stored in a single event queue along with their putative times, which may be at fixed times or drawn from a probability distribution. When an event is triggered, it may trigger the addition or removal of future events on the queue, or the modification of their rates, thus causing a recalculation of their putative time. This approach is adapted from the next-reaction method[60], which optimizes the Gillespie first-reaction method[61] by storing all events on an indexed binary heap. This data structure is simple to implement and sufficiently fast and compact to store all events in the system, down to individual state transitions for each infection course within each host. Specifically, modifying the putative time for an event on the queue is O(log $N$) and heap storage is O($N$), where $N$ is the number of events.

**Statistical analyses.** Selection vs. neutral models: In order to disentangle signatures of immune selection from those of transmission per se in parasite population structures, we designed neutral models in which hosts do not build specific immunity towards alleles or genes, in addition to the selection model described above. In the complete neutrality model (Fig. 1d), when hosts are infected, the duration of infection is determined by the deactivation rate of each gene, which follows a Poisson distribution of a constant rate; thus, hosts do not build immunity after an infection. The rate of deactivation is calculated to match the average duration of infection of the corresponding selection model, while maintaining the rest of the parameters (e.g., $G$, $b$) (Supplementary Fig. 1). In the generalized immunity model, the duration of infection decreases as the number of past infections increases, similar to the selection model. However, the identity of the alleles does not have a role. We therefore match the average curve of duration of infection vs. number of past infections to that of the corresponding selection scenario (See Fig. 1).

Diversity metrics, as well as epidemiological parameters, are calculated after each run to compare between scenarios (see Supplementary Note 1 and Supplementary Fig. 8). Diversity is quantified using common measures from ecology, including Shannon diversity[62] ($H = -\sum_{i=1}^{S} p \ln(p_i)$), Simpson's diversity and evenness[63] ($D_{simpson} = \frac{1}{\sum_{i=1}^{S} p_i^2}$, $E = S/\sum_{i=1}^{S} p_i^2$), β-diversity (i.e., turnover in composition of *var* genes or repertoires among parasite samples in time $\sum_{i}^{S}\left|x_{ij} - x_{ik}\right|/\sum_{i}^{S}\left|x_{ij} + x_{ik}\right|$), as well as within-repertoire diversity at the allelic and genetic levels. Within-repertoire diversity is calculated as the number of unique alleles or genes divided by the potential maximum number of unique alleles or genes (e.g., 60 genes and 120 alleles if the genome size is 60 and the number of epitopes is 2). EIR, prevalence, and MOI are also compared among model runs under different parameter settings and scenarios.

Building of similarity networks: In addition to diversity, similarity networks based on allelic composition at the gene or repertoire levels are built to investigate parasite population structure. For this purpose, 150 parasites are sampled at 120-day intervals in the hosts, to subsample the simulations in a way that is meaningful for later empirical application of network analyses. Directional similarity networks for *var* genes or parasite genomes (i.e., *var* repertoires) are built with edges

encoding the proportion of shared unique alleles. Specifically,

$$S_{ij} = \frac{a}{U_i}; S_{ji} = \frac{a}{U_j}$$

where $a$ is the shared number of unique alleles (or genes) between repertoires $i$ and $j$, and $U_i$ and $U_j$ are the total number of unique alleles (or genes) in repertoires $i$ and $j$, respectively. This directional index of genetic similarity is designed to represent the relative asymmetric competition between two repertoires, as explained in Supplementary Fig. 9.

Calculation of network properties: For the inspection of network structures from repertoires that have strong similarities, we retained edges with the top 1% of edge weights. Thirty-four network properties are calculated to detect selection signatures and to distinguish these from patterns generated by pure transmission dynamics or generalized immunity. These properties include metrics of transitivity, degree distributions, component sizes, diameters, reciprocity, and proportion of three-node graph motifs (see Supplementary Table 1 for a complete list of properties and definitions). For inspection of modular structures, an additional metric is introduced and named "module $F_{ST}$". This metric quantifies to what extent the strain modules inferred from repertoire similarity networks are genetically different from each other, by comparing the genetic diversity within and between modules[64,65].

Simulations and machine learning algorithms for classification: For each combination of parameters (i.e., initial gene pool size $G$, biting rate $b$, and duration of naive infection $D$), 100 simulations were run to calculate the distribution of the network properties under the scenarios of immune selection, complete neutrality, and generalized immunity. We investigated the accuracy of network classification under different quantiles of retained edges ranked by similarities. We explored the range of retained links from 80% to 10%. The properties are then transformed into non-correlated principle components. Discriminant analyses[66] are performed on the retained principle components that explain more than 90% of the variance, to design functions that maximize the differences among networks generated under different scenarios while minimizing the within-scenario variance. The accuracy of the discriminant functions is assessed by the proportion of correct classifications (i.e., power). Here we use the lower two posterior probabilities of classification assignment among the three scenarios as the false positive rates. A similar approach is followed to build a classifier for empirical networks. Details are given below.

**Comparisons with empirical data.** Data sampling: The empirical data analyzed here was collected from a study performed across two catchment areas in BD, Ghana located in the Upper East Region near the Burkina Faso border. Malaria in BD is hyperendemic and is characterized by marked seasonal transmission of *P. falciparum* during the wet season between June and October. This age-stratified serial cross-sectional study was conducted over two sequential seasons. The first survey was completed at the end of the wet season in October 2012, followed by a second survey at the end of the dry season between mid-May and June 2013. Details on the study population, data collection procedures and epidemiology have been published elsewhere[8]. Briefly, after obtaining informed consent, finger prick blood samples were collected for parasitological assessment for *P. falciparum* by blood smears and dried blood spots for molecular genotyping[8]. Sequencing was conducted for all isolates that are microscopically positive but asymptomatic, with no strong bias in sequencing parasite genomes across host ages. The study was reviewed and approved by the ethics committees at the Navrongo Health Research Center, Ghana; the Noguchi Memorial Institute for Medical Research, Ghana; New York University, USA; the University of Melbourne, Australia; and the University of Chicago, USA.

PCR amplification and *var* DBLα sequence analysis: The DBLα domain of *P. falciparum var* genes were amplified from genomic DNA using universal degenerate primers, as previously described[67]. Amplicons were pooled and barcoded libraries were sequenced on an Illumina MiSeq sequencer using the 2 × 300 paired-end cycle protocol, MiSeq Reagent kit v3 chemistry (NYUGTC, New York USA; AGRF, Melbourne, Australia). A custom pipeline was developed to de-multiplex and remove PCR and sequencing artefacts from the DBLα sequence tags. Reads were demultiplexed into individual fastq files for each isolate using flexbar v2.5[68] and paired based on valid combinations of molecular identifier (MID) tags in the forward and reverse reads. A minimum read length of 100nt and a maximum uncalled bases threshold of 15 were used. The resulting paired fastq files were then merged using PEAR v.0.9.10[69], to ensure the resulting merged fastq files had appropriate base quality scores allowing for filtering of low quality reads. The minimum assembly length was set to 100nt and the minimum overlap required between a read pair was set to 20nt. Low-quality reads were filtered if they had more than one expected error using the fastq_filter option of Usearch v8.1.1832[70]. Next, chimeras were filtered using Uchime denovo and then the filtered reads were clustered using the cluster_fast function of Usearch after the removal of singletons to reduce the impact of errors. A threshold of 96% identity[15] was used to cluster the reads. To increase the overall quality of the sequences, the resulting clusters were removed if they contained < 15 reads to remove low support reads. The representative read from each cluster was kept for the remaining stages of the pipeline. Next, any non-DBLα sequences were filtered out with a domain score threshold of 80. Finally, as a quality check the remaining reads were aligned to the reference *var* DBLα sequences of the 3D7, Dd2, and HB3 laboratory clones from

experimental sequence data. To subsequently determine DBLα types shared between isolates, the cleaned DBLα reads were clustered using a pipeline based on the USEARCH software suite version 8.1.1831[70]. Initially, duplicate reads were removed and the remaining reads were sorted by how many duplicates were present using the derep_prefix command. The remaining reads were then clustered at 96% pairwise identity using the usearch cluster_fast command. Finally, the original unfiltered reads were aligned back to the centroids of the clusters and an operational taxonomic unit table was generated using the usearch_global command before a binary version of the table was generated.

Building of empirical networks and model prediction: Empirical networks are built from var DBLα types sequenced and processed from 1284 P. falciparum isolates from individuals residing in BD, Ghana. Following a previously published analysis framework, the DBLα types are translated into all six reading frames and classified into either upsA or upsB/upsC (i.e., non-upsA) groups[67]. Gene networks are built based on pairwise similarities of unique upsB/upsC DBLα types that are above 0.6. The choice of the threshold is based on the average within-class sequence similarity of the 24 DBLα subclasses (see %ID in Figure 2 of Rask et al.[16]). As infections by multiple parasite genomes (MOI > 1) are very common in malaria endemic regions, we selected isolates with a total number of upsB/upsC DBLα types ranging from 40 to 55 copies, to maximize the probability of selecting hosts with single-clone infections, which reduced the number of isolates to 161 (see main text for rationale of focusing on upsB/upsC DBLα types; Supplementary Data 1). Because of the need to consider as many repertoires as possible in our network analyses, we considered all isolates regardless of their age. The repertoire similarity network is built among these isolates (Supplementary Fig. 5).

In order to build a classifier for the empirical network, a library of simulated networks was generated for parameter ranges representative of Ghana: global var gene pools from 10,000 to 20,000 (according to the sampling counts), duration of naive infection equal to 1 year, and mean biting rates ranging from 0.1 to 0.5 person per day. This selection of parameters results in EIR values similar to those reported for Upper East Region of Ghana in Appawu et al.[33] (see Supplementary Fig. 2). We generated a vector of monthly seasonal biting amplitudes relative to the mean biting rate, which resembles the local mosquito population of lowland in Appawu et al.[33]. The simulated networks are then constructed by sampling 161 random isolates from two periods 7 months apart from each other (the end of October and the beginning of June), resembling the sampling regime of the empirical data. The trained discriminant functions from simulated networks[66] are then applied to empirical networks to predict whether they are generated under a dominant role of immune selection versus that of neutrality. The Bayesian posterior probability of classification is calculated by assuming Gaussian densities of prior distributions of each class.

**Data availability**. The original C++ code for the var evolution model is available on Github (https://github.com/pascualgroup/VarModel). The python code for the sequence cleaning pipeline is available on GitHub at https://github.com/UniMelb-Day-Lab/DBLaCleaner. The python code to determine DBLα types is available on GitHub at https://github.com/UniMelb-Day-Lab/clusterDBLalpha. This Targeted Locus Study project has been deposited at DDBJ/ENA/GenBank under the Bio-Project Number: PRJNA 396962.

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

## Acknowledgements

This research was supported by the Fogarty International Center at the National Institutes of Health [Program on the Ecology and Evolution of Infectious Diseases (EEID), Grant number: R01-TW009670]. S.P. was supported by a James S. McDonnell Foundation 21st Century Science Initiative—Postdoctoral Program in Complexity Science-Complex Systems Fellowship Award and by a Fulbright Fellowship from the U.S. Department of State. We thank the participants, communities, and the Ghana Health Service in BD, Ghana, for their willingness to participate in this study. We also thank the personnel at the Navrongo Health Research Centre for sample collection and parasitological assessment/expertise. We are grateful to Abraham R. Oduro, Anita Ghansah, and Kwadwo Koram for their helpful input related to the field study, to Gerry Tonkin-Hill for the development of the Illumina sequencing, cleaning, and clustering pipelines, and to Michael F. Duffy and Andrew P. Dobson for their insightful comments on an earlier version of the manuscript. We appreciate the support of the University of Chicago through computational resources of the Midway cluster.

## Author contributions

Q.H., S.P., and M.P. conceived and designed the study. Y.A.R. and Q.H. worked on the model. E.B.B. and Q.H. wrote the simulation code. K.E.T. and K.P.D. designed and led the data collection in Ghana. K.E.T. and S.R.P. performed the molecular experiments and genetic sequencing. Q.H. and S.P. analyzed the data. Q.H., S.P., and M.P. wrote the paper. All authors contributed to the final version of the manuscript.

## Additional information

**Competing interests:** The authors declare no competing interests.

