## [Peer Review File · Nature Communications]

Reviewers' comments:

Reviewer #1 (Remarks to the Author):

This is a timely paper adapting methods and concepts from community ecology (niches, limiting similarity) and seeking to apply and test them in the context of strain theory and epidemiology.

The center of the analysis relates to the development of new null models appropriate in this context, and then showing that empirical data departs from these null models, and shows evidence for non-neutral structure appearing in the similarity of pathogens in empirical data. Specifically, clusters of similar strains appear in a way that is consistent with these clusters occupying separate niches, which in turn are limited in their similarity due to competition.

I have some comments and suggestions related to these null models and their application:

-- The neutral models introduced seem reasonable, but I think that they could be presented in more detail in the main body of the paper, perhaps if possible with a conceptual figure (perhaps an embellishment of Extended Data Figure 6). One particular issue that may benefit from parsing out more explicitly was competition in the completely neutral model—given that it was stated (L124) that repertoires in this model do not compete for hosts, I found it confusing that some repertoires could outcompete others (L175 and Fig 1 yellow).

Reading in more detail about the structure of the completely neutral model on L445, if I understood correctly repertoires in the completely neutral case can still differ due to the deactivation rates of each gene. So what I took from this was that there are fitness differences between repertoires, even in this neutral case. If this is the correct interpretation, it perhaps could be spelled out further in the main text. It also suggests a even “more” neutral case, where there are no fitness differences between repertoires (i.e. all genes have equal deactivation rates). Perhaps this is too extreme, and if so then that could be explained.

In short, a conceptual figure might convey precisely in what ways repertoires do and do not differ across each of these models.

-- Ecologists may be used to thinking about speciation events leading to novel species which still in some sense are similar to their parent species. This mode of evolution can therefore lead on its own to clusters in a space of traits, particularly if the trait space is very large, and the neutral system has not had time to explore this space fully. Some kind of discussion of how sensitive is the lack of clustering in the neutral scenarios to the details of how new repertoires are generated would be interesting to ecological readers. Clearly the recombinative process (which here is modeled as a random sampling of genes from two ‘parent’ strains) leads to variants exploring the space of possible variants much faster than a gradual random walk—but it would be interesting to quantify how critical this is for the (lack of) clustering.

-- It may help readers to see a little more detail on the application of Girvan-Newman algorithm and the resulting F_{st} between groups as a measure of limiting similarity, as stated in Fig 1. First, is there a threshold imposed on repertoire similarity to define links, or are some repertoires simply not overlapping at all? (If the latter, I would assume this may be somewhat sensitive to the number of genes in a repertoire, which may in effect be providing an implicit threshold). If there is a threshold, how sensitive are the resulting groupings to this? I would also like to see a clearer comparison with quantification of limiting similarity in ecology. (The authors cite reference 29 as an example of quantifying limiting similarity in community ecology, in Extended Table 2.)

-- I understand the approach taken in Fig 3 puts Ghana in the regime of immune selection. It would also be informative to show the structure of this empirical network explicitly, in the same terms as Figure 1.

Reviewer #2 (Remarks to the Author):

The authors examine diversity in the antigenic repertoire of *Plasmodium falciparum*, developing a novel framework to determine whether the patterns of diversity are consistent with selection by host immunity on specific epitopes. They use an individual-based model to identify metrics most capable of distinguishing among the scenarios of immune selection on specific var gene repertoires, the impact of general (i.e., non-strain specific) immunity, and a null model where strains infect and persist independent of their var gene repertoires and previous exposure on the part of the host. Finally, they simulate these three models with parameters relevant to transmission intensity in Ghana, finding that the observed data correspond most closely to the scenario of immune selection on specific repertoires.

The manuscript is compelling and well-written, but I have a few concerns.

1. Does it make sense to assume that the duration of infections decreases as the number of infections increases in both the generalized immunity and selection models? Nassir et al. 2005 *Int J Parasitol* found a positive correlation between MOI and persistence within the host, and that pattern can be explained theoretically (Klein et al. 2014 *Inf and Imm*). Is there any way that pattern could emerge from the generalized immunity or neutral models considered in this manuscript?

2. Can the authors comment on previous work reporting a signature of immune selection for rare surface variants (including Bull et al. 1999 *Infection and Immunity*, Fig. 4)? Are those previous methods prone to error/less robust? The methods used in the present manuscript are quite complicated, and it would be good to make the case that such complexity is required.

3. Felger et al. 2012 (*PLoS ONE*) find an empirical pattern of relatively steady force of infection and infection duration across host ages, but differences in parasite detectability. The Ghana data appear to include host age (line 499), but could the authors clarify how (or whether) they addressed host age in the analysis?

4. The references to control (e.g., line 274-6) seem like vague afterthoughts that either need to be fleshed out or omitted. Are the authors envisioning targeting specific PfEMP1

variants or repertoires with emerging technology, or targeting the diversity indirectly by changing transmission intensity (e.g., by scaling up vector control efforts and/or drug treatment)?

5. The Ghana data set consists of samples taken from two different times, either at the end of the wet season (so that hosts will have been exposed to a large number of strains) or at the end of the dry season (where presumably the number of strains within the host has been gradually declining due to competition, immunity, stochastic loss, etc.). I can imagine that these two sets of surveys would have different expected patterns based on the neutral, generalized immunity and immune selection models—were those different expectations accounted for in some way or are the data lumped together for comparisons?

Minor points:

1. Line 48-49: What is meant by the phrase 'multi-genome *P. falciparum* isolates'?
2. The key assumptions of the general immunity model could be explained more clearly—for example, in line 121, the consequences for individual hosts are mentioned but not the critical epidemiological parameters. What does generalized immunity mean for infection duration? I think the answer is in the paragraph on comparing neutral versus selection models (beginning on line 441), but the wording is vague. For the generalized immunity model, "the duration of infection decreases as the number of infections increases". Would that be the number of active infections currently within the host or the number of infections triggered in that host up until the current time point?
3. For hosts infected with multiple strains, the per-strain transmission probability is 'p' divided by the number of strains (lines 369-371), so the statement in line 418 that "the host is considered infections with the active strain" is confusing—infectious here presumably means a nonzero probability of transmission? The transmission probability is also defined as $1/(\text{number of repertoires})$ in line 764—could it be made clear that definition is the same as the first one?
4. What is the biological rationale/meaning of lines 423-424 that after "the gene is deactivated, the host gains immunity to all the new epitopes"?
5. The figure captions are not as informative as they could be. The figures would be easier to interpret if the first bolded statement of the caption specified the key pattern in the figure.
6. Line 568: It would be good to mention the language the code is in.
7. Fig. 2a, b—the key is very far from the key title (Percentage of correct classification of selection model), and the key is very close to the x-axis label for duration. Further, 'Percentage' must actually be 'Proportion', or else the model does a very poor job of correctly classifying the selection model.
8. I realize space is very limited but some of the supplemental figures belong in the main text. I would prefer to see Extended Data Fig. 1 in the main text, since the model structure is central to the argument and the figure makes the important assumptions more accessible to a broad audience. Fig. 3 in the main text would have more impact if the corresponding results for the simulations were also in the main text. Fig. 3a mirrors Extended Data Fig. 4b-d, and the meaning would be more obvious if those panels were in the main text. Likewise the scatterplots most likely to reflect the Ghana data could be plotted in the main text (Fig. 2h perhaps?). Some of the panels in the main text could be moved to the supplement if need be, including Fig. 1b & e, Fig. 2a & e (and potentially some of the

scatter plots). To be clear, the panels illustrating model validation are important (e.g., the figures showing that F_{st} values cannot distinguish between the different cases), but they could be moved to the supplement.

9. Could in-out degree distribution be clearly defined for non-specialists?

Reviewer #3 (Remarks to the Author):

This paper presents an interesting and rather complex study of the competition dynamics emerging at the within-host and between-host levels between polymorphic malaria parasites. Using a combination of simulated and empirical networks of antigenic diversity, the paper develops a framework for disentangling neutral from non-neutral structuring processes, e.g. immune selection acting on var gene repertoires. The authors show that at least in a few cases, for example, at high transmission intensities, there are distinct features of genetic similarity networks between sampled host repertoires, by which one can distinguish immune selection dynamics from generalized immunity and total neutrality. The final part of the paper is application of this network classification tool to one var gene dataset from Ghana to conclude that these data are more consistent with immune selection processes rather than neutral ones. Although I like the paper overall and I believe it brings forth new insights into the epidemiology and population structure of *P. falciparum*, I think there are many issues to address and clarify before the paper is accepted for publication.

1. First the premise of the paper is that these three processes (immune selection, generalized immunity and neutral dynamics) are mutually exclusive. While I do not share this view, I suggest that the authors present a more nuanced discussion of how these three processes (and many more, including host population structure/spatial and non-spatial heterogeneity) could be acting in parallel to shape strain structure in *P. falciparum*. In this respect, I would also suggest a reworking of the title: for example: "Networks of genetic similarity can reveal how neutral and non-neutral processes shape strain structure in *P. falciparum*." Since the algorithm correctly classifies these scenarios only for special combinations of transmission intensity and duration of infection (Figure 2) it is important to state, how do these parameter combinations compare to typical endemic scenarios of malaria, namely how relevant and widely applicable could this algorithm be in general.

2. I believe the authors could also add in their introduction references to other pathogen systems where questions of neutral vs. non-neutral forces shaping their epidemiological dynamics and population structure have been and are being investigated, to place their study in a wider context beyond malaria parasites. To mention just a few examples: Lipsitch et al. *Epidemics*. 2009 Mar;1(1):2-13. doi: 10.1016/j.epidem.2008.07.001, Nicoli et al *Theor Popul Biol*. 2015 Aug;103:84-92. doi: 10.1016/j.tpb.2015.05.003, Gjini, & Madec, *Theoretical Ecology*, 2017 Mar, 10(1) 129-141, Cobey & Lipsitch *Science*. 2012 Mar 16;335(6074):1376-80.

3. The introduction, in my view, is also missing a concise presentation of the novelty of this study. Compared to an earlier study by some of the authors, published in eLife, there are many similarities in model formulation and analysis. What are the main novelties here, in

the method, in the question, and the advances in biological understanding compared to the previous work? In particular in line 206, the authors could say they extended a previous model instead of 'developing a model' which gives the impression that this is all new. Please highlight the key novelties appropriately.

4. The paper does not contain a clear reference to the relationship between diversity of var gene repertoires and prevalence. Did the authors impose a fixed prevalence to their simulations and data in a top-down manner? Would they expect a different prevalence under their three models? Can the structural differences between these models and their immunological feedbacks produce a different relationship between R_0 and prevalence? A discussion of this relationship is extremely important when it comes to targeting diversity for control, as the authors state in the end of their abstract.

5. The authors use the term diversity confusingly throughout the paper, sometimes to refer to the var gene pool size and the dimensionality of the system (e.g. line 210, line 255), sometimes to refer to actual comparison between genes or repertoires. I would suggest them to check carefully all the instances where this is used and to be consistent. In Statistical analyses, the exact formulae of each diversity index should be presented to facilitate understanding by the reader. The authors should also read and refer to an important paper in the field of diversity indices, by Leinster & Cobbold *Ecology* 2012 Mar;93(3):477-89, arguing on the importance of species similarity in diversity measures and putting forward a comparison on the basis of diversity profiles.

6. The authors refer to 'strength of competition' at various places in the manuscript. I think this is misleading, because what is being varied is typically a parameter of transmission intensity that the authors take as a 'proxy' for competition. The emergent competition between repertoires and var genes is a complex nonlinear higher order property of the system. There is an indirect relationship between transmission intensity, and all other model parameters, and the magnitude of 'epidemiological competition' experienced by the parasite strains, and this is what the paper is about. So I would ask the authors to use the direct parameter that they varied, instead of 'competition', for example in Figure 1, Extended Data Figure 3, but also in other instances throughout the paper, e.g. lines 208-210.

7. The hypothesis of the paper outlined in lines 144-154 seems to miss reference to the importance of recombination processes in shifting the balance between signatures of selection and neutral forces. The last sentence of this paragraph is not immediately obvious. This would likely depend on the magnitude of recombination rate, which on one hand acts to homogenize repertoires and on the other creates diversity. As transmission increases, competition between repertoires increases, because hosts get more and more exposed and build immunity, but also mixing between repertoires via recombination increases, and which one dominates will depend on the rates of these processes.

8. There need to be more details about how the simulations and sampling of repertoires were performed to ensure reproducibility of analyses. In particular:
a. In Figure 1: 1 node=1 repertoire=1 host with multiplicity of infection 1? What is the total host population size? What is the prevalence of infection in this population and how does it

change over time? What percentage of the total prevalence does the sample size of 150 individuals represent? Why is only the top 1% of the edges used in the analysis?

b. Can the authors comment on the size of the modules or module size distribution as possible discriminant feature between models? It seems that there is a strong signature for many small modules in the immune selection scenario, as opposed to fewer larger modules in the other two cases.

c. How do the network properties change over time in all scenarios? How long does it take for each of them to reach equilibrium? Is it true that all network properties settle at some equilibrium value? Or if not, is it the ones that show fewer fluctuations that are able to distinguish the three epidemiological macro scenarios?

d. In figure 2: What is the endemic prevalence of infection resulting from each combination of biting rate and infection duration? For each parameter combination how many 'epidemiological dynamics' were run? How many hosts sampled in how many time-points? Was there any seasonality implemented in the biting rate? How was this sampling time-point selected? Again in this figure, the authors use interchangeably competition and transmission intensity, and this should be avoided. It seems at higher transmission intensities, there is more information on the competitive feedbacks between strains mediated by immune selection. And this information becomes more apparent in the network measures used. Is subfigure e suggesting that in all three instances the classification accuracy is similar? It does not seem to be supported by subfigure f, where all the points appear to overlap. Please clarify.

e. In Methods, how is N_s , the number of strains transmitted per donor determined? What is the value of P_c (line 393) this parameter does not appear in the Table of parameters, neither parameter τ .

f. I am confused about the evolutionary dynamics, namely via ectopic recombination and mutation. How is the diversity between genes determined? First in the initial time point when the gene pool size is initiated and then later on? Equation 2 in line 396 describes the dependence on the similarity of parental genes, how is this determined in these simulations? Each allele is assumed different from all the others and mutations occurring as in the infinite allele model? So similarity between two var genes is just the number of identical epitope alleles that they share (0/1)? Perhaps some graphic illustration would help to visualize and understand better the measures of diversity at each level and the processes responsible.

g. Can a host be co-infected with two strains upon the same transmission event? In that case which strain precedes the other in order of expression of var genes, or are they expressed simultaneously? How is immunity to each strain acquired in that case? Sequential or in parallel?

h. In Statistical analyses, the authors state they calculate the metrics after each run. When does each run finish? How is that determined?

9. Related to figure 2 again. Are there any network properties that could discriminate between models at lower transmission intensities? It seems that these parameter regimes are more problematic. If not, what alternative ways could there be to increase the applicability of the classification algorithm for low endemicity scenarios? Any alternative sampling schemes?

10. How does the rate of waning of immunity affect the discrimination between the models? Could one use the rate of waning of immunity as a tune-up and tune-down parameter for interpolating between neutral (instantaneous waning) and immune selection (very slow waning) scenarios?

11. The authors should provide more details about the classification algorithm applied to the empirical network constructed with the Ghana dataset. Which parameters were used in the simulations of each model? Are there any references to support the assumed values of these parameters? How robust are the final classification probabilities to slight variations in parameter values, e.g. rate of waning of immunity? Please also specify regarding the LD components used for the final classification. what proportion of variance between networks do they capture?

12. Since the authors already have a full simulation tool with individual-level detail both for current carriage and history of past exposures, could they comment on the possibility to use the network of 'host immune history' similarity as opposed to 'current infecting parasite strain similarity' as a means to detect signatures of selection vs. neutral processes?

Response to reviewers' comments

Reviewer #1 (Remarks to the Author):

This is a timely paper adapting methods and concepts from community ecology (niches, limiting similarity) and seeking to apply and test them in the context of strain theory and epidemiology.

The center of the analysis relates to the development of new null models appropriate in this context, and then showing that empirical data departs from these null models, and shows evidence for non-neutral structure appearing in the similarity of pathogens in empirical data. Specifically, clusters of similar strains appear in a way that is consistent with these clusters occupying separate niches, which in turn are limited in their similarity due to competition.

I have some comments and suggestions related to these null models and their application:

1. -- The neutral models introduced seem reasonable, but I think that they could be presented in more detail in the main body of the paper, perhaps if possible with a conceptual figure (perhaps an embellishment of Extended Data Figure 6). One particular issue that may benefit from parsing out more explicitly was competition in the completely neutral model—given that it was stated (L124) that repertoires in this model to not compete for hosts, I found it confusing that some repertoires could outcompete others (L175 and Fig 1 yellow).

Reading in more detail about the structure of the completely neutral model on L445, if I understood correctly repertoires in the completely neutral case can still differ due to the deactivation rates of each gene. So what I took from this was that there are fitness differences between repertoires, even in this neutral case. If this is the correct interpretation, it perhaps could be spelled out further in the main text. It also suggests a even “more” neutral case, where there are no fitness differences between repertoires (i.e. all genes have equal deactivation rates). Perhaps this is too extreme, and if so then that could be explained.

In short, a conceptual figure might convey precisely in what ways repertoires do and do not differ across each of these models.

Response: We thank the referee for pointing out that this important aspect of the paper was confusing as presented. We agree that the current description of the two neutral models and the immune selection model in the main text should be more detailed. We have now added to the description of the models in the Methods section as well as in Results section 1, and specifically stated that there are no fitness differences between strains in the neutral models. We also added model comparisons to Figure 1d and Supplementary Figure 1, to make the differences among the three models clearer to the reader.

Strains in the completely neutral model do not differ in their deactivation rate for each gene within repertoires (this rate is a constant rate with a Poisson distribution). Because the fitness of a strain is defined by how long they persist in a host, there is no fitness difference between strains in the two neutral models. However, fitness can differ in the immune selection model, because the identities of genes do matter given the memory that a host acquires of previous genes it has encountered. Genes that have been seen are not expressed, which shortens the time of infection and therefore decreases the effective transmission rate and the fitness of the strain. In the network analysis, we used network measures that can differentiate among the scenario in which competition matters and those in which it does not. The asymmetrical measure of similarity between the two repertoires is designed to measure competition and potential fitness differences resulting from immune selection. In particular, we observe more single directional links under neutral models, exactly because neutral models do not select for all the genes to be unique in the repertoire. We therefore changed the wording on L175 and figure 1 legend to clarify these elements of the models and of the network measures.

We changed Line 175 to: In the two null models, repertoires with lower number of unique genes are not removed by selection, and therefore, when one repertoire consists of a higher number of unique epitopes than the other, there can be only one directed edge between the pair because the similarity of the edge in the opposite direction is weak.

2. Ecologists may be used to thinking about speciation events leading to novel species which still in some sense are similar to their parent species. This mode of evolution can therefore lead on its own to clusters in a space of traits, particularly if the trait space is very large, and the neutral system has not had time to explore this space fully. Some kind of discussion of how sensitive is the lack of clustering in the neutral scenarios to the details of how new repertoires are generated would be interesting to ecological readers. Clearly the recombinative process (which here is modeled as a random sampling of genes from

two 'parent' strains) leads to variants exploring the space of possible variants much faster than a gradual random walk—but it would be interesting to quantify how critical this is for the (lack of) clustering.

Response: The recombination processes (both the shuffling of parental genes in the offspring repertoire, as well as ectopic recombination in the asexual stage) occur in both the neutral and immune selection scenarios at the same rate. New recombinant repertoires are indeed analogous to new phenotypes in trait space in ecology, and can create some clustering. However, the rate of removal of new recombinant repertoires (or creation of clusters in trait space) differs due to the presence of immune selection in the full model, which leads to a clearer cluster structure in this selection case. We showed in Figure 3a (version 1 Figure 1a) that in low transmission scenarios (whose chance of recombination is lower due to the reduced possibility of multiplicity of infection), the immune selection scenario is harder to differentiate from the neutral ones. We now include material on the rate of recombination, how it is accelerated by competition, and its impact on the immune selection signatures in the discussion, line 376-386.

3. It may help readers to see a little more detail on the application of Girvan-Newman algorithm and the resulting F_{st} between groups as a measure of limiting similarity, as stated in Fig 1. First, is there a threshold imposed on repertoire similarity to define links, or are some repertoires simply not overlapping at all? (If the latter, I would assume this may be somewhat sensitive to the number of genes in a repertoire, which may in effect be providing an implicit threshold). If there is a threshold, how sensitive are the resulting groupings to this? I would also like to see a clearer comparison with quantification of limiting similarity in ecology. (The authors cite reference 29 as an example of quantifying limiting similarity in community ecology, in Extended Table 2.)

Response: We defined links based on a cut-off for edge weights and inspected structures based on the top 1% strongest links. Modularity is computed for the networks with these links. By contrast, F_{st} uses all the similarity information of edge weights and therefore doesn't involve a cutoff in the edge weights. The modules defined to calculate pairwise F_{st} are based on the GN algorithm applied to the network with only strong links to reduce noise-to-data ratio. We now include a new table that shows the sweeping of the cutoff and the corresponding accuracy in network differentiation using DAPC. We found that as long as cutoffs are higher than 20% in the quantile of edge weights, they do not change the overall pattern of the corresponding accuracy in network differentiation for simulated cases (Supplementary Table 2), despite important changes in network features.

We included discussion of the comparison of limiting similarity with the network measures in the discussion in line 338-345.

4. I understand the approach taken in Fig 3 puts Ghana in the regime of immune selection. It would also be informative to show the structure of this empirical network explicitly, in the same terms as Figure 1.

Response: we do show this structure in the Supplementary Figure 6 (version1: Extended data figure 5).

Reviewer #2 (Remarks to the Author):

The authors examine diversity in the antigenic repertoire of *Plasmodium falciparum*, developing a novel framework to determine whether the patterns of diversity are consistent with selection by host immunity on specific epitopes. They use an individual-based model to identify metrics most capable of distinguishing among the scenarios of immune selection on specific var gene repertoires, the impact of general (i.e., non-strain specific) immunity, and a null model where strains infect and persist independent of their var gene repertoires and previous exposure on the part of the host. Finally, they simulate these three models with parameters relevant to transmission intensity in Ghana, finding that the observed data correspond most closely to the scenario of immune selection on specific repertoires.

The manuscript is compelling and well-written, but I have a few concerns.

Response: We thank the referee for the positive evaluation of our work, and for the raised concerns which have pointed us to several confusing explanations and missing information in our previous text.

1. Does it make sense to assume that the duration of infections decreases as the number of infections increases in both the generalized immunity and selection models? Nassir et al. 2005 Int J Parasitol found a positive correlation between MOI and persistence within the host, and that pattern can be explained theoretically (Klein et al. 2014 Inf and Imm). Is there any way that pattern could emerge from the generalized immunity or neutral models considered in this manuscript?

Response: There is a general trend for the duration of infection decreasing as the number of previous infections (not the number of concurrent infections, i.e., MOI) increases, in the variant-specific immune selection model. There is also substantial variance around this trend, simply because the hosts only build variant specific immunity, and will still have long infections when infected with strains with new variants (see Supplementary Figure 1). We used the average duration per number of previous infections from the

immune selection model, and applied this quantity to the duration of infection in the corresponding generalized immunity model to create comparable duration profiles between the two models for meaningful comparison (see Supplementary Figure 1).

For our main results, we do not have a cost associated with MOI, since we consider that each parasite repertoire expresses var types and proliferates independently. We did explore another version of within-host dynamics, in which the switching rate of var genes is reduced as a function of MOI, to mimic the observation that each strain dominates at a different time of the infection due to expression of a different var. All the strains within a multiply infected host persist longer under this assumption (as seen in Nassir et al. 2005 *Int J Parasitol* and Bruce et. al 2000 *Parasitology*; which also resembles the model prediction in Klein et al. 2014). The network structure remains qualitatively the same. We have included these modifications and results in the discussion and corresponding supplementary information (Supplementary Fig. 2).

2. Can the authors comment on previous work reporting a signature of immune selection for rare surface variants (including Bull et al. 1999 *Infection and Immunity*, Fig. 4)? Are those previous methods prone to error/less robust? The methods used in the present manuscript are quite complicated, and it would be good to make the case that such complexity is required.

Response: Bull et al. 1999 Fig. 4 showed results of agglutination experiments demonstrating that children who have broader protection against PfEMP1 variants tend to be infected by only rare variants. We think these previous methods are robust despite their small sample size. Our method inspects immune selection at two distinct hierarchical levels: those of repertoires and genes respectively, with an emphasis on the structure at the repertoire level, which is that of primary interest for strain structure. Thus, we specifically used network methods to identify a role of immune selection at the repertoire level. This is novel as this level has not been addressed by previous methods. For completeness, we also showed an immune selection signature at the gene level based on evidence that common variant types are genetically more distinct from other genes. Our findings and those of Bull et al. 1999 address different and complementary aspects of immune selection: Bull et al. 1999 showed from the hosts' perspective that those individuals who have built more immunity can only be infected by rare (presumably newer) variants; we showed that from the parasite's perspective, common variants can only persist if they are genetically distinct. Our main question is about structure at the repertoire level or 'strain' structure, as the effect of immune

selection at the genes' level is already recognized (the high diversity of the var genes is an indication of this process; so are the findings by Bull et al.). This effect at the genes' level does not necessarily imply however that immune selection is able to structure the parasite population, especially given the large gene pool and the high recombination rates. We have made this distinction and question clearer now in the Intro and Discussion.

Our methods might appear complicated because they are based on network analyses of complex data, but all the patterns have clear biological and interpretable meaning. We also think that networks provide a natural construct to address population structure in genetic systems with recombination. They are the meaningful alternative to trees, which would not apply. Our analysis relies on a standard set of network properties, except for the definition of modules' F_{st} .

3. Felger et al. 2012 (PLoS ONE) find an empirical pattern of relatively steady force of infection and infection duration across host ages, but differences in parasite detectability. The Ghana data appear to include host age (line 499), but could the authors clarify how (or whether) they addressed host age in the analysis?

Response: In the agent-based model, we fitted an exponential distribution to the reported Bongo population demography (provided by Tiedje K.) and estimated an average life span of 30 years. The force of infection is kept the same across host age in the model, and infection duration is not varied a priori as a function of age but determined by whether a given host has seen the particular strain in the past. Felger et al. 2012 find that infection duration is highest in 5-9 year old children, who are more likely to be infected by completely new strains. This is consistent with our model. In our empirical data, parasite var repertoires are analyzed by PCR, and sequencing is conducted for all isolates that are microscopically positive but asymptomatic. Therefore, there should not be a strong bias in sequencing parasite genomes across host ages. Because of the need to consider as many repertoires as possible in our network analyses, these were conducted by considering samples regardless of age. The development and testing of the methods based on the theory relies on a similar sampling of the simulated 'data'.

4. The references to control (e.g., line 274-6) seem like vague afterthoughts that either need to be fleshed out or omitted. Are the authors envisioning targeting specific PfEMP1 variants or repertoires with

emerging technology, or targeting the diversity indirectly by changing transmission intensity (e.g., by scaling up vector control efforts and/or drug treatment)?

Response: We thank the referee for pointing out the weakness of our closing statement. We have now deleted the closing sentence, and replace it by more specific writing. We are largely thinking of the indirect effects on diversity and on the importance of monitoring these changes, and better understanding its consequences for the resilience of the system to elimination efforts.

5. The Ghana data set consists of samples taken from two different times, either at the end of the wet season (so that hosts will have been exposed to a large number of strains) or at the end of the dry season (where presumably the number of strains within the host has been gradually declining due to competition, immunity, stochastic loss, etc.). I can imagine that these two sets of surveys would have different expected patterns based on the neutral, generalized immunity and immune selection models—were those different expectations accounted for in some way or are the data lumped together for comparisons?

Response: We pooled the single infections from the two seasons mainly to achieve a larger sample size (since there aren't sufficient single infections in the separate seasons) to perform robust testing of network structure. Although repertoires are sampled from two seasons, the overall distributions of PTS and var frequencies remain constant across these time points (Ruybal-Pesante et al. in prep), which suggests a stationary structure. Since the strains sampled in the dry season are presumably from infections continued from the wet season, they would represent samples from the same parasite population.

As suggested, we have extended our analyses to consider model simulations that explicitly include seasonality, and found that the classification results do hold, with the empirical network still classified as more likely to be generated by the immune selection scenario (see details in Results: empirical data comparison and discussion).

Minor points:

1. Line 48-49: What is meant by the phrase 'multi-genome *P. falciparum* isolates'?

Response: This refers to isolates consisting of multiple repertoires based on their multiplicity of infection. We changed this phrase to "Multiplicity of infection [MOI] >1" and carefully explained in the text the meaning of isolates vs. repertoires on line 286-287 (Result section: comparison with empirical data): An isolate refers to a complete sample of parasites from a host, which may contain multiple infections (MOI>1).

2. The key assumptions of the general immunity model could be explained more clearly—for example, in line 121, the consequences for individual hosts are mentioned but not the critical epidemiological parameters. What does generalized immunity mean for infection duration? I think the answer is in the paragraph on comparing neutral versus selection models (beginning on line 441), but the wording is vague. For the generalized immunity model, “the duration of infection decreases as the number of infections increases”. Would that be the number of active infections currently within the host or the number of infections triggered in that host up until the current time point?

Response: The number of infections here means the past number of infections the host has experienced. The decrease in average duration of infection with the number of past infections in a given host for the general immunity model is matched with the duration for a similar number of previous infections on average from the immune selection model. We have clarified this assumption in lines 144-148, 155-158 in the methods, along with the supplementary figure 1. A more detailed explanation can also be found in the response to major point 1.

3. For hosts infected with multiple strains, the per-strain transmission probability is ‘p’ divided by the number of strains (lines 369-371), so the statement in line 418 that “the host is considered infectious with the active strain” is confusing—infectious here presumably means a nonzero probability of transmission? The transmission probability is also defined as $1/(\text{number of repertoires})$ in line 764—could it be made clear that definition is the same as the first one?

Response: We assume that infections from different repertoires in the host are independent from each other. But the repertoire is not infectious until gametocytes are transmitted and oocysts develop in the mosquito for the sexual stage, and in the initial liver stage for the host. Once the repertoire starts to express var genes in the blood stage, it is considered infectious. When the host is co-infected by multiple repertoires, each repertoire has a chance equal to $1/(\text{number of repertoires})$ to be transmitted from the host to the mosquito. This part of the methods has been rewritten to clarify the transmission rules better.

4. What is the biological rationale/meaning of lines 423-424 that after “the gene is deactivated, the host gains immunity to all the new epitopes”?

Response: When one gene is deactivated and another gene is activated, the host includes the deactivated gene variant into their immunity memory (this sentence has been rewritten as such in the methods).

5. The figure captions are not as informative as they could be. The figures would be easier to interpret if the first bolded statement of the caption specified the key pattern in the figure.

Response: we have followed the suggestion and summarized the key pattern in the first bolded statement of the figure legends.

6. Line 568: It would be good to mention the language the code is in.

Response: We have added the language of the code in the data availability section.

7. Fig. 2a, b—the key is very far from the key title (Percentage of correct classification of selection model), and the key is very close to the x-axis label for duration. Further, ‘Percentage’ must actually be ‘Proportion’, or else the model does a very poor job of correctly classifying the selection model.

Response: we have changed ‘percentage’ to ‘proportion’, and readjusted key positions

8. I realize space is very limited but some of the supplemental figures belong in the main text. I would prefer to see Extended Data Fig. 1 in the main text, since the model structure is central to the argument and the figure makes the important assumptions more accessible to a broad audience. Fig. 3 in the main text would have more impact if the corresponding results for the simulations were also in the main text. Fig. 3a mirrors Extended Data Fig. 4b-d, and the meaning would be more obvious if those panels were in the main text. Likewise the scatterplots most likely to reflect the Ghana data could be plotted in the main text (Fig. 2h perhaps?). Some of the panels in the main text could be moved to the supplement if need be, including Fig. 1b & e, Fig. 2a & e (and potentially some of the scatter plots). To be clear, the panels illustrating model validation are important (e.g., the figures showing that F_{st} values cannot distinguish between the different cases), but they could be moved the supplement.

Response: We agree. As suggested, we moved Extended Data Fig. 1, 4 to the main text.

9. Could in-out degree distribution be clearly defined for non-specialists?

Response: As suggested, we now more clearly defined in the text in-edges indicate that the focal node can be outcompeted by the other node, whereas out-edges indicate the opposite, with the focal node outcompeted by the other node. Thus, in-degree and out- measure respectively the number of in-edges and out-edges for a focal node. The biological meaning of directional edges are better explained in the response to Reviewer 1 point 1, as well as in the caption of supplementary Fig. 4.

Reviewer #3 (Remarks to the Author):

This paper presents an interesting and rather complex study of the competition dynamics emerging at the within-host and between-host levels between polymorphic malaria parasites. Using a combination of simulated and empirical networks of antigenic diversity, the paper develops a framework for disentangling neutral from non-neutral structuring processes, e.g. immune selection acting on var gene repertoires. The authors show that at least in a few cases, for example, at high transmission intensities, there are distinct features of genetic similarity networks between sampled host repertoires, by which one can distinguish immune selection dynamics from generalized immunity and total neutrality. The final part of the paper is application of this network classification tool to one var gene dataset from Ghana to conclude that these data are more consistent with immune selection processes rather than neutral ones. Although I like the paper overall and I believe it brings forth new insights into the epidemiology and population structure of *P. falciparum*, I think there are many issues to address and clarify before the paper is accepted for publication.

1. First the premise of the paper is that these three processes (immune selection, generalized immunity and neutral dynamics) are mutually exclusive. While I do not share this view, I suggest that the authors present a more nuanced discussion of how these three processes (and many more, including host population structure/spatial and non-spatial heterogeneity) could be acting in parallel to shape strain structure in *P. falciparum*. In this respect, I would also suggest a reworking of the title: for example: “Networks of genetic similarity can reveal how neutral and non-neutral processes shape strain structure in *P. falciparum*.” Since the algorithm correctly classifies these scenarios only for special combinations of transmission intensity and duration of infection (Figure 2) it is important to state, how do these parameter combinations compare to typical endemic scenarios of malaria, namely how relevant and widely applicable could this algorithm be in general.

Response: We thank the referee for raising up this important issue as it is one we do not wish to be misunderstood. It is not the premise of our paper that these three processes are mutually exclusive. In fact, generalized immunity must be operating in malaria dynamics, and this process would arise from immunity to more conserved antigens. The main question we are addressing is whether a distinct

population structure exists of var repertoires which reflects an important role of immune selection. Because the theoretical expectations of previous strain theory are not sufficient to guide our analyses of the data, we formulated theory intended to identify signatures of immune selection. This requires comparisons to null models, and the neutral models are intended as such. They allow us to define and identify patterns that can be solely generated by demography and transmission (i.e., the complete neutrality model), as well as generalized immunity (a more realistic neutral model that is in fact the implicit assumption in many epidemiological models of malaria), and to distinguish those patterns from those generated under variant-specific immunity. We have now added text in the Introduction and Discussion to make this clear and present the work in the more nuanced way the referee mentions.

We note that by using network properties, we can differentiate the signatures with high or reasonable accuracy for most combinations of parameter settings, except for low diversity and low transmission. This makes sense because when there is limited diversity, and little competition due to low transmission, immune selection is expected to be much less important in these transmission regimes. Thus, we expect the approach to be widely applicable. In the specific application to Ghana data, we have selected parameter ranges that encompass values from the literature for high endemic regions (see lines 304-305). The other range combinations considered are also related to specific geographic regions (see lines 198-199) for medium diversity.

2. I believe the authors could also add in their introduction references to other pathogen systems where questions of neutral vs. non-neutral forces shaping their epidemiological dynamics and population structure have been and are being investigated, to place their study in a wider context beyond malaria parasites. To mention just a few examples: Lipsitch et al. *Epidemics*. 2009 Mar;1(1):2-13. doi: 10.1016/j.epidem.2008.07.001, Nicoli et al *Theor Popul Biol*. 2015 Aug;103:84-92. doi: 10.1016/j.tpb.2015.05.003, Gjini, & Madec, *Theoretical Ecology*, 2017 Mar, 10(1) 129-141, Cobey & Lipsitch *Science*. 2012 Mar 16;335(6074):1376-80.

Response: We thank the referee for these references, which have now been added (line 368-374) to place the work in a wider context.

3. The introduction, in my view, is also missing a concise presentation of the novelty of this study. Compared to an earlier study by some of the authors, published in eLife, there are many similarities in model formulation and analysis. What are the main novelties here, in the method, in the question, and the advances in biological understanding compared to the previous work? In particular in line 206, the authors could say they extended a previous model instead of ‘developing a model’ which gives the impression that this is all new. Please highlight the key novelties appropriately.

Response: We agree with the referee that the novelty of the work needs to be made clearer earlier on, and included in the Introduction. We had previously included this information but later in the text, and as such, the writing was not effective. We have now added text in the Introduction to make clear earlier in the manuscript what is novel in this work and model. The referee is correct, we have extended the model, but this involves some important modifications that allow us to (1) consider an open system, where new genes can be constantly added, and (2) match the vast gene diversity of empirical systems. We also built, and considered comparisons to null models. Consideration of such neutral (dynamical) models was absent from previous work. In fact we have shown that networks with clear modules (clear non-overlapping clusters of repertoires as predicted by early theory) are not necessarily indicative of the existence of immune selection. Low diversity and low transmission can also produce similar patterns under neutral models (See network structures in Extended Data Figure 2, low and medium competition). With no explicit comparison to such null models, we therefore cannot conclude that immune selection is responsible for the existing population structure just by observing non-overlapping niches/modules. In brief, the PNAS publication (Day et al. 2017) showed the existence of a *non-random* population structure of limited overlap, and the eLife one (Artzy et al., 2012) mainly extended the original strain theory to multi-copy gene families. This work is the first presentation of (1) theory that applies to the vast combinatorial complexity of empirical systems at high endemism, (2) methods to identify non-neutral population structure at high endemism, and (3) conclusive evidence for a role of immune selection in the non-random structure discovered earlier. We hope these contributions are now clearer in the revised text.

4. The paper does not contain a clear reference to the relationship between diversity of var gene repertoires and prevalence. Did the authors impose a fixed prevalence to their simulations and data in a top-down manner? Would they expect a different prevalence under their three models? Can the structural differences between these models and their immunological feedbacks produce a different relationship

between R_0 and prevalence? A discussion of this relationship is extremely important when it comes to targeting diversity for control, as the authors state in the end of their abstract.

Response: Prevalence in the model is an emergent property, as is also Entomological Inoculation Rate (EIR). We now include a supplementary figure 2 that displays the relationship between prevalence and EIR in our model under different diversity and transmission intensities for the three models. Although prevalence increases with EIR under certain biting rates, the prevalence values are higher for complete neutrality model than for immune selection or generalized immunity, given the same parameter combinations. This is because in the completely neutral model, there is no differential distribution of infections among different age classes (an unrealistic feature resulting from, each host having a fixed rate of clearing parasites, which does not decrease with a higher number of infections). In the generalized and immune selection cases, infections have a similar age pattern; therefore the EIR and prevalence relationships are very similar between the two cases. We agree with the importance of the relationship between diversity and prevalence in terms of disease control. We have now added results on this in lines 158-164.

5. The authors use the term diversity confusingly throughout the paper, sometimes to refer to the var gene pool size and the dimensionality of the system (e.g. line 210, line 255), sometimes to refer to actual comparison between genes or repertoires. I would suggest them to check carefully all the instances where this is used and to be consistent. In Statistical analyses, the exact formulae of each diversity index should be presented to facilitate understanding by the reader. The authors should also read and refer to an important paper in the field of diversity indices, by Leinster & Cobbold *Ecology* 2012 Mar;93(3):477-89, arguing on the importance of species similarity in diversity measures and putting forward a comparison on the basis of diversity profiles.

Response: Although we agree with the referee on the importance of a careful use of the term diversity, as far as we can tell for the two cases mentioned (e.g. line 210, line 255), diversity refers consistently to the same property, that is, the var gene pool size. We have now carefully checked the text to clarify the usage of diversity: we changed all the instances of “high/medium diversity gene pool” to “high/medium gene pool size”; diversity was clearly defined in other instances (repertoire diversity, or the system dimensionality); In the paragraph that calculates ecological diversity metrics, we cited the papers of the specific calculations we use. As suggested, we added the formula in the method section as well. We

recognize the importance of diversity profiles proposed in Leinster & Cobbold (2012) by considering species similarity in diversity measures. We did not use this measure because epitopes are considered independent in our model, thus diversity at the allelic level can use the traditional diversity measures we proposed (Supplementary Fig. 9). Diversity structure at the genetic and repertoire level is studied using network properties instead of diversity indices. We cited the study in the text and included it in our supplementary table 3 when discussing network approach comparisons with the quantification of limiting similarity in ecology.

6. The authors refer to ‘strength of competition’ at various places in the manuscript. I think this is misleading, because what is being varied is typically a parameter of transmission intensity that the authors take as a ‘proxy’ for competition. The emergent competition between repertoires and var genes is a complex nonlinear higher order property of the system. There is an indirect relationship between transmission intensity, and all other model parameters, and the magnitude of ‘epidemiological competition’ experienced by the parasite strains, and this is what the paper is about. So I would ask the authors to use the direct parameter that they varied, instead of ‘competition’ , for example in Figure 1, Extended Data Figure 3, but also in other instances throughout the paper, e.g. lines 208-210.

Response: We agree that the strength of competition is a complex nonlinear property of the system, even though it should be an increasing function of transmission intensity We have changed the instances where usage of “transmission intensity” is more appropriate. We only refer to competition when explaining the underlying mechanisms of the patterns per se.

7. The hypothesis of the paper outlined in lines 144-154 seems to miss reference to the importance of recombination processes in shifting the balance between signatures of selection and neutral forces. The last sentence of this paragraph is not immediately obvious. This would likely depend on the magnitude of recombination rate, which on one hand acts to homogenize repertoires and on the other creates diversity. As transmission increases, competition between repertoires increases, because hosts get more and more exposed and build immunity, but also mixing between repertoires via recombination increases, and which one dominates will depend on the rates of these processes.

Response: As transmission intensity increases, the mixing between repertoires via recombination increases, which creates many unfit repertoires, and also results into higher competition selecting for

more unique repertoires. These two forces should synergistically increase the importance of immune selection. It is also true that ectopic recombination during the asexual stage creates new variant types, which are preferred in the immune selection scenario. Although both processes are termed recombination, they are caused by completely different mechanisms, and have almost opposite effects in the case of immune selection. We have clarified and added a paragraph with discussion of the different recombination processes (Lines 376-386).

8. There need to be more details about how the simulations and sampling of repertoires were performed to ensure reproducibility of analyses. In particular:

a. In Figure 1: 1 node=1 repertoire=1 host with multiplicity of infection 1? What is the total host population size? What is the prevalence of infection in this population and how does it change over time? What percentage of the total prevalence does the sample size of 150 individuals represent? Why is only the top 1% of the edges used in the analysis?

Response: We thank the referee for requesting these clarifications. In the simulations, each node is a randomly sampled parasite repertoire, which might exist as a single infection in a host or as one of the infections in a host. We used this random sampling scheme because we are interested in the parasite population structure. The percentage of nodes that are from MOI=1 hosts varies based on the specific parameter settings. Prevalence and EIR are now provided on Supplementary Fig. 2. Host population size is 10000. Prevalence of infection does not change after the simulation reaches a stationary state. The prevalence of the 150 samples would change depending on the different prevalence of a specific run (Supplementary Fig. 2). We only considered the strongest edges in the network to study the highest overlap among repertoires, which are most likely to reveal structure. As suggested, we have included a new figure that shows the sweeping of the cutoff and the influence of this cut-off on the accuracy in network differentiation using DAPC. (See lines 247-255 and Supplementary table 2).

b. Can the authors comment on the size of the modules or module size distribution as possible discriminant feature between models? It seems that there is a strong signature for many small modules in the immune selection scenario, as opposed to fewer larger modules in the other two cases.

Response: These two features are included in Supplementary Table 1, 13 and 14.

c. How do the network properties change over time in all scenarios? How long does it take for each of them to reach equilibrium? Is it true that all network properties settle at some equilibrium value? Or if

not, is it the ones that show fewer fluctuations that are able to distinguish the three epidemiological macro scenarios?

Response: The system always reaches a stationary state and the distribution of the properties do not fluctuate over time. The temporal structure of the network properties is a very interesting question worth investigating, but beyond the scope of the current study. Work is underway in that direction as now indicated in the Discussion. We note that consideration of network structure over time requires even the definition of a different (multilayer) network and additional properties. Since the regions of interest experience endemic malaria, the transient dynamics (initial stage depending on initial conditions) of the simulations before these reach a stationary state, are not relevant epidemiologically. They would be if one were to address responses to control implementations, which is also an interesting subject deserving investigation in a separate analysis.

We therefore only inspected network properties after the system reaches stationarity (about 70 years, as the expected host age in the model is ~ 30 years). We then calculated network properties averaged across 10 time slices and 100 independent runs. Network properties still fluctuate among runs, but within a certain range for each scenario/parameter settings. It is not the case however that properties with the least fluctuations are those best able to distinguish among the scenarios. What matters is whether their distributions or joint distributions among the three scenarios have the largest disparity. In addition, combinations of network properties rather than individual network properties have much stronger power of differentiation among the three scenarios, especially for high gene pool size cases (Fig. 3b)

d. In figure 2: What is the endemic prevalence of infection resulting from each combination of biting rate and infection duration? We have now added this information in supplementary figure 2. For each parameter combination how many ‘epidemiological dynamics’ were run? 100 realizations. How many hosts sampled in how many time-points? 150 parasites from randomly sampled hosts in 10 distinct time slices. Was there any seasonality implemented in the biting rate? We had not implement seasonality in the initial investigation of structure differences among the three models. We have now followed the initial presentation of results for the non-seasonal case, with the seasonal case in relation to the empirical comparison with Ghana data. (Fig. 5b). How was this sampling time-point selected? We selected a time point once the transients of the system have died out to consider the stationary state only (ie no

dependence on initial conditions). We considered several time slices per network, separated by at least 10 years apart to remove temporal correlations among different layers.

Again in this figure, the authors use interchangeably competition and transmission intensity, and this should be avoided. It seems at higher transmission intensities, there is more information on the competitive feedbacks between strains mediated by immune selection. And this information becomes more apparent in the network measures used. Is subfigure e suggesting that in all three instances the classification accuracy is similar? It does not seem to be supported by subfigure f, where all the points appear to overlap. Please clarify.

Response: figure 2e only shows the proportion of correct classifications for the selection case. It is true that the classification of neutral and generalized immunity is not high because the network structures resemble each other. We are not interested in differentiating between these two neutral cases, and it is interesting that they generate similar patterns, as generalized immunity is biologically more realistic, whereas complete neutrality is really intended as a theoretical construct, which results in some emergent patterns that are not epidemiologically meaningful (such as a uniform age distribution). In subgraph 2f, we only showed for the purpose of illustration one pair of network features. Many more features are actually considered together to classify the scenarios much more accurately. We have now changed figure 2f to present the contribution of important network features in the model classifications.

e. In Methods, how is N_s , the number of strains transmitted per donor determined? What is the value of P_c (line 393) this parameter does not appear in the Table of parameters, neither parameter τ .

Response: The same number of strains that the mosquito has acquired from the donor will be transmitted to the receiver. P_c is the proportion of ectopic recombination that results in conversion (alleles being converted into one parental gene) versus recombination (reciprocal change). In the current implementation, we kept all the ectopic recombination resulting into reciprocal change. We therefore removed this parameter in the revision. We set τ (recombination tolerance) to 0.8 as suggested from experiments in Drummond et al (2005) PNAS.

f. I am confused about the evolutionary dynamics, namely via ectopic recombination and mutation. How is the diversity between genes determined? First in the initial time point when the gene pool size is initiated and then later on? Equation 2 in line 396 describes the dependence on the similarity of parental

genes, how is this determined in these simulations? Each allele is assumed different from all the others and mutations occurring as in the infinite allele model? So similarity between two var genes is just the number of identical epitope alleles that they share (0/1)? Perhaps some graphic illustration would help to visualize and understand better the measures of diversity at each level and the processes responsible.

Response: Genes are composed of linear epitopes and a genome/repertoire is a combination of different genes (Figure 1). For the epitopes, we use an infinite allele model to generate new alleles (i.e., a new mutation always adds a new allele to the pool). However, new alleles can be lost and die out by stochasticity. The referee is correct: the similarity between genes in the model is computed as the number of identical epitope alleles. We first generate a gene pool that is a random combination of epitope alleles. During transmission, these genes within the genome can experience ectopic recombination (and therefore, exchange alleles), and the new gene might be functional depending on the probability calculated using Eq. 2. We have now illustrated and explained these different aspects of the model better in the Figure 1 caption and methods.

g. Can a host be co-infected with two strains upon the same transmission event? In that case which strain precedes the other in order of expression of var genes, or are they expressed simultaneously? How is immunity to each strain acquired in that case? Sequential or in parallel?

Response: A host can be co-infected with multiple strains in the same transmission event. In the current implementation, co-infected strains act independently from each other (parallel expression). They switch var genes independently at their own rate based on host immunity. The host gains immunity after the parasite switches to a different var gene. We did explore another version of within-host dynamics, in which the switching rate of var genes is reduced as a function of MOI, to mimic the observation that each strain dominates at a different time due to expression of a different var. All the strains within a multiply infected host persist longer under this assumption (as seen in Nassir et al. 2005 Int J Parasitol and Bruce et. al 2000 Parasitology; which also resembles the model prediction in Klein et al. 2014). The network structure remains qualitatively the same. We have included these modifications and results in the supplementary information and discussion (Supplementary Fig. 7).

h. In Statistical analyses, the authors state they calculate the metrics after each run. When does each run finish? How is that determined?

Response: We run each simulation for 100 years, and sample the networks after 70 years as explained earlier to be sure transients have been removed and only the stationary state is sampled. We sample enough individual networks to obtain a representative distribution of network properties.

9. Related to figure 2 again. Are there any network properties that could discriminate between models at lower transmission intensities? It seems that these parameter regimes are more problematic. If not, what alternative ways could there be to increase the applicability of the classification algorithm for low endemicity scenarios? Any alternative sampling schemes?

Response: we argued that for low endemicity scenarios, the importance of immune selection will be less, as expected, and therefore, the selection scenario should be harder to distinguish from the neutral ones. . But we also show in Fig. 2e that we can still differentiate between selection and neutrality if the diversity of var genes is high relative to the transmission rate. We do not seek to increase the applicability of the classification because the inability to differentiate among these underlying mechanisms reflects a real lack of importance of selection (ie the weakness of this force). For all analyses, we need the ensemble of properties.

10. How does the rate of waning of immunity affect the discrimination between the models? Could one use the rate of waning of immunity as a tune-up and tune-down parameter for interpolating between neutral (instantaneous waning) and immune selection (very slow waning) scenarios?

Response: The rate of waning of immunity is based on experimental evidence testing acute vs. covalent serums, which is around 1/100 days (Collins et al. 1968, Krause et al. 2007, Infect. Immun.). We did not vary the rate in the model, but longer immune memory would impose stronger immune selection. It would be sensible to use this parameter's variation to interpolate complete neutrality and immune selection, but not generalized immunity. Because tuning up and down waning immunity does not create the age structure that generalized immunity achieves, and would like to compare in our study.

11. The authors should provide more details about the classification algorithm applied to the empirical network constructed with the Ghana dataset. Which parameters were used in the simulations of each model? Are there any references to support the assumed values of these parameters? How robust are the final classification probabilities to slight variations in parameter values, e.g. rate of waning of immunity?

Please also specify regarding the LD components used for the final classification. what proportion of variance between networks do they capture?

Response: Parameters were varied within a range rather than applying point estimates (Lines 660-663). Specifically, we vary var gene pool size from 10,000 to 20,000 based on local var type counts of empirical data we used. We lack empirical estimates for rates of mosquitoes taking consecutive bites of humans, but the range of values was chosen to generate prevalence and EIR values that are comparable to those reported from the field work at these sites documented in Tiedje et al. 2017 and Appawu et al. 2014 Tropical Medicine & International Health. Therefore, the ranges we applied should be realistic for the high endemic scenario of the Ghana site. Our analyses should be robust to variations in parameters, since we did not rely on narrow point values. We did not vary the rate of waning immunity in our simulations. However, we varied biting rate and gene pool size to capture the empirical range of EIR observed in this region (Appawu et al. 2014 Tropical Medicine & International Health). The LD functions are ordered by their eigenvalues, which measure the proportion of the variance of the original data each function captures. We included the values on the figure and provided explanation in the caption.

12. Since the authors already have a full simulation tool with individual-level detail both for current carriage and history of past exposures, could they comment on the possibility to use the network of ‘host immune history’ similarity as opposed to ‘current infecting parasite strain similarity’ as a means to detect signatures of selection vs. neutral processes?

Response: This is an intriguing idea. We have explored the network of host immune history briefly. The host immune history network exhibits however a very strong age-structure, which makes it more difficult to detect signatures of selection at the strain level. We see that older hosts have seen a higher number of var types, as expected. It is also harder to rely on this network to consider empirical data because we do not have yet hosts’ serum data to do immune agglutination experiments. Signatures of selection at the repertoire level are therefore most easily detected through parasite genome similarity patterns directly as we have done.

Reviewers' comments:

Reviewer #1 (Remarks to the Author):

This revised paper adapts methods and concepts from community ecology (niches, limiting similarity) and seeks to apply and test them in the context of strain theory and epidemiology.

I thank the authors for the clarifying comments in their response, and for the corresponding changes in the text and figure additions/reordering.

I have some remaining comments.

** It may help readers to see a little more detail on the application of Girvan-Newman algorithm and the resulting F_{st} between groups as a measure of limiting similarity, as stated in Fig 1. First, is there a threshold imposed on repertoire similarity to define links, or are some repertoires simply not overlapping at all? (If the latter, I would assume this may be somewhat sensitive to the number of genes in a repertoire, which may in effect be providing an implicit threshold). If there is a threshold, how sensitive are the resulting groupings to this? I would also like to see a clearer comparison with quantification of limiting similarity in ecology. (The authors cite reference 29 as an example of quantifying limiting similarity in community ecology, in Extended Table 2.)

"Response: We defined links based on a cut-off for edge weights and inspected structures based on the top 1% strongest links. Modularity is computed for the networks with these links. By contrast, F_{st} uses all the similarity information of edge weights and therefore doesn't involve a cutoff in the edge weights. The modules defined to calculate pairwise F_{st} are based on the GN algorithm applied to the network with only strong links to reduce noise-to-data ratio. We now include a new table that shows the sweeping of the cutoff and the corresponding accuracy in network differentiation using DAPC. We found that as long as cutoffs are higher than 20% in the quantile of edge weights, they do not change the overall pattern of the corresponding accuracy in network differentiation for simulated cases (Supplementary Table 2), despite important changes in network features."

— Thank-you for these clarifications and additional analyses. I still think that the measures of limiting similarity could possibly be linked more tightly to existing methods in the ecological literature. The authors cite classic work introducing the concept of limiting similarity, where at late times single species emerge in separate niches. What the authors are computing is in contrast a measure of clustering, which has since been established is a natural outcome when a source of novelty is combined with competitive interactions (e.g. Scheffer, Marten, and Egbert H. van Nes. "Self-organized similarity, the evolutionary emergence of groups of similar species." *Proceedings of the National Academy of Sciences* 103.16 (2006): 6230-6235.).

Other recent work has established various metrics for characterizing this clustering (e.g. the distribution of distances from a modal abundant species, in Jeraldo, Patricio, et al.

"Quantification of the relative roles of niche and neutral processes in structuring gastrointestinal microbiomes." *Proceedings of the National Academy of Sciences* 109.25 (2012): 9692-9698, or k-means dispersion in Competition and immigration lead to clusters of similar species, not trait separation

Rafael D'Andrea Rocha, Maria Riolo, Annette Marie Ostling, bioRxiv 264606; doi: <https://doi.org/10.1101/264606>). Ecologically-focused readers may appreciate the comparison with (or at least discussion of) these existing approaches to test for clustering.

**I understand the approach taken in Fig 3 puts Ghana in the regime of immune selection. It would also be informative to show the structure of this empirical network explicitly, in the same terms as Figure 1.

"Response: we do show this structure in the Supplementary Figure 6 (version1: Extended data figure 5)."

— What I was missing was the statistical summary in the right-hand column of Figure 2 (version 1 Figure 1) demonstrating the distribution of motifs (etc) in the empirical data. Is it possible/meaningful to visualize these in the same way as in the current Figure 2?

Reviewer #2 (Remarks to the Author):

The authors have addressed my major concerns, and I have a few other minor points I would like to call to their attention.

1. In their reply to my earlier comments, the authors state

In the agent-based model, we fitted an exponential distribution to the reported Bongo population demography (provided by Tiedje K.) and estimated an average life span of 30 years. The force of infection is kept the same across host age in the model, and infection duration is not varied a priori as a function of age but determined by whether a given host has seen the particular strain in the past. Felger et al. 2012 find that infection duration is highest in 5-9 year old children, who are more likely to be infected by completely new strains. This is consistent with our model. In our empirical data, parasite var repertoires are analyzed by PCR, and sequencing is conducted for all isolates that are microscopically positive but asymptomatic. Therefore, there should not be a strong bias in sequencing parasite genomes across host ages. Because of the need to consider as many repertoires as possible in our network analyses, these were conducted by considering samples regardless of age. The development and testing of the methods based on the theory relies on a similar sampling of the simulated 'data'.

I have no problem with this approach but this reply contains important methodological details that should be presented somewhere in either the main text or the supplement. I don't see anyone being able to replicate this study without this information.

2. Lines 94-94: This sentence is pretty cryptic—rephrase for better comprehensibility?

3. Line 265-266: Could the authors add (brief) text to explain why seasonal networks are able to distinguish between generalized immunity and complete neutrality when non-seasonal networks cannot?
4. Line 315: I leave it up to the authors but 'unequivocal' might not stand the test of time, and I don't think it's necessary.

Typos:

Line 149: Sentences shouldn't begin with parameter symbols.

Fig. 1c is referred to before 1b—reorder panel labeling?

Line 261: "transmissions" should be "transmission"

Line 368: Italicize 'Pertussis'

Line 1102: {REF} missing.

Reviewer #3 (Remarks to the Author):

I think the authors have done a great job in improving and clarifying their manuscript. They have adequately responded to all of my comments and included supplementary modifications and results that make their study and methods more accessible and reproducible.

I now feel that the paper is suitable for publication, making clear and novel contributions, both at theoretical and practical level, to epidemiological strain theory and to understanding malaria dynamics.

Reviewers' comments:

Reviewer #1 (Remarks to the Author):

This revised paper adapts methods and concepts from community ecology (niches, limiting similarity) and seeks to apply and test them in the context of strain theory and epidemiology.

I thank the authors for the clarifying comments in their response, and for the corresponding changes in the text and figure additions/reordering.

I have some remaining comments.

** It may help readers to see a little more detail on the application of Girvan-Newman algorithm and the resulting F_{st} between groups as a measure of limiting similarity, as stated in Fig 1. First, is there a threshold imposed on repertoire similarity to define links, or are some repertoires simply not overlapping at all? (If the latter, I would assume this may be somewhat sensitive to the number of genes in a repertoire, which may in effect be providing an implicit threshold). If there is a threshold, how sensitive are the resulting groupings to this? I would also like to see a clearer comparison with quantification of limiting similarity in ecology. (The authors cite reference 29 as an example of quantifying limiting similarity in community ecology, in Extended Table 2.)

"Response: We defined links based on a cut-off for edge weights and inspected structures based on the top 1% strongest links. Modularity is computed for the networks with these links. By contrast, F_{st} uses all the similarity information of edge weights and therefore doesn't involve a cutoff in the edge weights. The modules defined to calculate pairwise F_{st} are based on the GN algorithm applied to the network with only strong links to reduce noise-to-data ratio. We now include a new table that shows the sweeping of the cutoff and the corresponding accuracy in network differentiation using DAPC. We found that as long as cutoffs are higher than 20% in the quantile of edge weights, they do not change the overall pattern of the corresponding accuracy in network differentiation for simulated cases (Supplementary Table 2), despite important changes in network features."

— Thank-you for these clarifications and additional analyses. I still think that the measures of limiting similarity could possibly be linked more tightly to existing methods in the ecological literature. The authors cite classic work introducing the concept of limiting similarity, where at late times single species emerge in separate niches. What the authors are computing is in contrast a measure of clustering, which has since been established is a natural outcome when a source of novelty is combined with competitive interactions (e.g. Scheffer, Marten, and Egbert H. van Nes. "Self-organized similarity, the evolutionary emergence of groups of similar species." *Proceedings of the National Academy of Sciences* 103.16 (2006): 6230-6235.).

Other recent work has established various metrics for characterizing this clustering (e.g. the distribution of distances from a modal abundant species, in Jeraldo, Patricio, et al.

"Quantification of the relative roles of niche and neutral processes in structuring gastrointestinal microbiomes." *Proceedings of the National Academy of Sciences* 109.25 (2012): 9692-9698, or k-means dispersion in Competition and immigration lead to clusters of similar species, not trait separation
Rafael D'Andrea Rocha, Maria Riolo, Annette Marie Ostling, bioRxiv 264606; doi: <https://doi.org/10.1101/264606>). Ecologically-focused readers may appreciate the comparison with (or at least discussion of) these existing approaches to test for clustering.

Response: We appreciate the reviewer's suggestion to better connect our findings to the recent advances in community ecology. We now expanded our introduction (Lines 56-58, 119-124) and discussion (Lines 359-380) to include recent findings on the evolutionary emergence of clusters of highly similar species under competitive interactions, which can be stabilized by immigration or evolution. We explained how such findings are linked to ours. We noted that the findings resemble our results for the low to medium diversity gene pool scenario, consistent with the effect of balancing selection in evolutionary genetics. Importantly, however, under a high diversity of the gene pool, clustering is no longer a characteristic pattern as we have described in our results; therefore, clustering measures (e.g., modularity and associated mFst) alone cannot distinguish immune selection from either of the neutral scenarios. The importance of the dimensionality of trait space underlying competition (and evolution) to the emergence of clusters (of species or strains) vs. more complex patterns of limiting similarity is now further emphasized in the Discussion in the context of the papers the referee has mentioned.

We have also included the clustering measures proposed by Rocha et al. 2018 in Table S3, which compares similar approaches in the three fields. We did not include the H statistics from Jeraldo et al. 2012 because it is well-known as the simplest measure of pairwise genetic distances, and it is not typically used in malaria studies (where PTS is the main metric).

**I understand the approach taken in Fig 3 puts Ghana in the regime of immune selection. It would also be informative to show the structure of this empirical network explicitly, in the same terms as Figure 1.

"Response: we do show this structure in the Supplementary Figure 6 (version1: Extended data figure 5)."

— What I was missing was the statistical summary in the right-hand column of Figure 2 (version 1 Figure 1) demonstrating the distribution of motifs (etc) in the empirical data. Is it possible/meaningful to visualize these in the same way as in the current Figure 2?

Response: We now included the comparisons of motif proportions in Supplementary Fig. S5 for the top 1% and 10% edges.

Reviewer #2 (Remarks to the Author):

The authors have addressed my major concerns, and I have a few other minor points I would like to call to their attention.

1. In their reply to my earlier comments, the authors state

In the agent-based model, we fitted an exponential distribution to the reported Bongo population demography (provided by Tiedje K.) and estimated an average life span of 30 years. The force of infection is kept the same across host age in the model, and infection duration is not varied a priori as a function of age but determined by whether a given host has seen the particular strain in the past. Felger et al. 2012 find that infection duration is highest in 5-9 year old children, who are more likely to be infected by completely new strains. This is consistent with our model. In our empirical data, parasite var repertoires are analyzed by PCR, and sequencing is conducted for all isolates that are microscopically positive but asymptomatic. Therefore, there should not be a strong bias in sequencing parasite genomes across host ages. Because of the need to consider as many repertoires as possible in our network analyses, these were conducted by considering samples regardless of age.

The development and testing of the methods based on the theory relies on a similar sampling of the simulated 'data'.

I have no problem with this approach but this reply contains important methodological details that should be presented somewhere in either the main text or the supplement. I don't see anyone being able to replicate this study without this information.

Response: We thank the reviewer for pointing out the need to clarify and include this information. We have now added this in appropriate places in the Methods (Lines 596-598, 662-664, 755-756, 805-807).

2. Lines 94-94: This sentence is pretty cryptic—rephrase for better comprehensibility?

Response: We agree. We have now removed this sentence as we found it was essentially redundant with what we have in the following paragraph. We have also extended the following paragraph which closes the Introduction. We believe the text is now clearer.

3. Line 265-266: Could the authors add (brief) text to explain why seasonal networks are able to distinguish between generalized immunity and complete neutrality when non-seasonal networks cannot?

Response: We are able to distinguish generalized immunity and complete neutrality in seasonal networks mainly due to the distribution of the duration of infection (see Fig. S1). Specifically, under generalized immunity, some infections (strains) are persistent enough to overcome the bottleneck in transmission during the dry season, and are therefore carried on into the wet season. In contrast, infections under complete neutrality all have relatively short durations that limit persistence across the dry season and to the next wet season. Therefore, the composition of strains under complete neutrality has a faster turnover rate than under generalized immunity. These different dynamics leave different signatures in the networks, captured by various network properties (e.g. in both motifs and diameter). In non-seasonal networks, where there are no bottlenecks for transmission, the influence of the duration of infection on strain diversity structure is less pronounced.

4. Line 315: I leave it up to the authors but ‘unequivocal’ might not stand the test of time, and I don’t think it’s necessary.

Response: we have changed the word to “clear”.

Typos:

Line 149: Sentences shouldn’t begin with parameter symbols.

Response: we rewrote the beginning of the sentence.

Fig. 1c is referred to before 1b—reorder panel labeling?

Response: we changed the figure ordering as the referee suggested.

Line 261: “transmissions” should be “transmission”

Response: we have changed to “transmission”.

Line 368: Italicize ‘Pertussis’

Response: we have italicized the word.

Line 1102: {REF} missing.

Response: we have fixed the reference.

Reviewer #3 (Remarks to the Author):

I think the authors have done a great job in improving and clarifying their manuscript. They have adequately responded to all of my comments and included supplementary modifications and results that make their study and methods more accessible and reproducible.

I now feel that the paper is suitable for publication, making clear and novel contributions, both at theoretical and practical level, to epidemiological strain theory and to understanding malaria dynamics.

Response: We thank the referee for the encouraging comments.

REVIEWERS' COMMENTS:

Reviewer #1 (Remarks to the Author):

I thank the authors for their expanded descriptions and analyses in response to my remaining comments.